# Evaluation Metrics in the Era of GPT-4: Reliably Evaluating Large Language Models on Sequence to Sequence Tasks

**Andrea Sottana**[1]   **Bin Liang**[1]   **Kai Zou**[1]   **Zheng Yuan**[2,1]

[1]NetMind.AI

[2]Department of Informatics, King's College London

{andrea.sottana, bin.liang, kz}@netmind.ai

zheng.yuan@kcl.ac.uk

## Abstract

Large Language Models (LLMs) evaluation is a patchy and inconsistent landscape, and it is becoming clear that the quality of automatic evaluation metrics is not keeping up with the pace of development of generative models. We aim to improve the understanding of current models' performance by providing a preliminary and hybrid evaluation on a range of open and closed-source generative LLMs on three NLP benchmarks: text summarisation, text simplification and grammatical error correction (GEC), using both automatic and human evaluation. We also explore the potential of the recently released GPT-4 to act as an evaluator. We find that ChatGPT consistently outperforms many other popular models according to human reviewers on the majority of metrics, while scoring much more poorly when using classic automatic evaluation metrics. We also find that human reviewers rate the gold reference as much worse than the best models' outputs, indicating the poor quality of many popular benchmarks. Finally, we find that GPT-4 is capable of ranking models' outputs in a way which aligns reasonably closely to human judgement despite task-specific variations, with a lower alignment in the GEC task.

## 1   Introduction

In recent years, Large Language Models (LLMs), particularly Transformer based (Vaswani et al., 2017; Devlin et al., 2019), have shown remarkable abilities across a wide range of NLP tasks. With the recent advances in capabilities of general-purpose generative models (Brown et al., 2020; Touvron et al., 2023), a range of NLP tasks can be reformulated as generation tasks.

Robust evaluation is still an unsolved problem and established automatic evaluation metrics have been found to be poor surrogates, correlating weakly with human judgement (Coyne et al., 2023). There is often no clear consensus on how these models should be evaluated (Mousavi et al.,

2022). Human evaluation has often been considered as the trusted evaluation method, though issues with human evaluation have also been widely acknowledged (Iskender et al., 2021), e.g. it can be difficult to reproduce (Cumbicus-Pineda et al., 2021). Nonetheless, a human evaluation study remains one of the best tools to sensibly assess any bias or limitation with automatic metrics (Liang et al., 2022).

Recent evaluation work has often focused on a single task (Zhang et al., 2023; Coyne et al., 2023), a single model (Bang et al., 2023), a single dataset (Gilardi et al., 2023) or automatic evaluation (Liang et al., 2022). In this work, we carry out a multi-dataset, multi-model, multi-task hybrid evaluation using automatic metrics, human evaluation, and model-to-model evaluation with GPT-4 (OpenAI, 2023).[1] We explore the open and closed-source LLMs space to sample the current landscape of available models and evaluate them on the following sequence-to-sequence tasks, reframed as text generation tasks without the requirement for task-specific fine-tuning: text summarisation, text simplification, and grammatical error correction (GEC).

These are our main findings: firstly, we show how traditional reference-based evaluation metrics are inadequate at predicting or replacing human judgement. It is unclear whether this is due to the limitations of the metrics or to the poor quality of references of large open source datasets, or both. While automatic metrics might have been an adequate proxy to evaluate previous models, they seem unable to reliably capture the performance of latest-generation LLMs which now generate ac-

---

[1]When preparing the manuscript, the authors have noticed that some recent work has also explored model-to-model evaluation, e.g. Chiang and Lee (2023); Liu et al. (2023); Fu et al. (2023). This paper makes a significant contribution by using an extensive set of metrics to provide a comprehensive evaluation of each model on several different sequence-to-sequence tasks.

ceptable output that is significantly different from the gold reference. Secondly, we prove that even open-source models outperform the gold standard reference of large and well-established datasets according to human evaluators. This shows how data quality is now one of the main bottlenecks in evaluation research. Finally, we reveal how GPT-4 has reasonable alignment with human judgement when ranking different models on most tasks and metrics; we did however observe some variations, with lower alignment in some metrics than in others. Our code is available at `https://github.com/protagolabs/seq2seq_llm_evaluation`.

## 2 Experimental Setup

### 2.1 Datasets

For text simplification, we used the Newsela test set (Xu et al., 2015), in particular the version used by Jiang et al. (2020). We randomly selected 3,000 samples after removing samples redundancy.[2] For text summarisation, experiments were run on 3,000 random samples taken from the CNN / DailyMail test set (Hermann et al., 2015; Nallapati et al., 2016). For GEC, we used the BEA-2019 Shared Task (Bryant et al., 2019) development set comprising of 4,384 samples.[3]

### 2.2 Models

All experiments were performed on a zero-shot unsupervised basis, without any additional fine-tuning or in-context learning, using a range of open-source LLMs and OpenAI commercial models.[4] We experimented with the HuggingFace[5] implementation of the following open-source models:[6] **Flan-T5** (google/flan-t5-xxl) (Chung et al., 2022); **T0pp** (bigscience/T0pp) (Sanh et al., 2021); **OPT-IML** (facebook/opt-iml-max-30b) (Iyer et al., 2022); **Flan-UL2** (google/flan-ul2) (Tay et al., 2022). The OpenAI models we used were **GPT-3** (text-davinci-003) (Brown et al., 2020); **InstructGPT** (davinci-instruct-beta) (Ouyang et al., 2022) and **ChatGPT**[7] (gpt-3.5-turbo-0301). For implementation

---

[2] More details are given in Appendix A.

[3] We did not use the test set as it was not fully disclosed.

[4] It is worth noting that some of the models used are already fine-tuned to follow instructions on a wide range of NLP tasks, some of which include the tasks above.

[5] http://huggingface.co

[6] The open-source models were run on local servers with up to 6 NVIDIA GeForce RTX 3090 GPUs each.

[7] https://openai.com/chatgpt

details, prompt engineering and hyper-parameter tuning, refer to appendix B.

## 3 Evaluation Metrics

We analysed models' outputs using both automatic metrics and human evaluation, and assessed the ability of the recently released GPT-4 model to act as a reviewer.

### 3.1 Automatic Evaluation

We used the most widely adopted reference-based metrics for each of the tasks. For text simplification, we report the SARI score (Xu et al., 2016). For text summarisation, we report the ROUGE score (Lin, 2004); following Phang et al. (2022), we compute the geometric mean of ROUGE-{1, 2, L} F1 scores. For GEC, we report the $F_{0.5}$ score computed using the ERRANT toolkit (Bryant et al., 2017).

### 3.2 Human Evaluation

Due to budgetary and time constraints, we recruited 3 human reviewers[8] through the Prolific platform[9] and asked them to review the quality of the models' outputs, as well as the gold reference on 100 randomly selected samples per dataset. All three reviewers were asked to annotate the same 100 samples for each of the three tasks. The studies were conducted on a customised version of the open-source POTATO annotation tool (Pei et al., 2022). For human evaluation of text summarisation, we followed the evaluation criteria and their definitions as adopted in Fabbri et al. (2021): *Relevance*, *Fluency*, *Coherence* and *Consistency*, on a 5-point Likert scale (Likert, 1932) from 1 to 5. For text simplification, we followed the evaluation criteria and their definitions as adopted in Grabar and Saggion (2022): *Semantics*, *Fluency* and *Simplicity*, on a 5-point Likert scale. For GEC, we adopted the *Over-correction* criterion from Fang et al. (2023) and introduced two new criteria: *Semantics* and *Grammaticality*. The definitions and assessment scales for these GEC criteria are detailed in Appendix C. The full set of instructions given to human reviewers for all tasks can be found in our GitHub repository linked above.

---

[8] We only accepted reviewers based in the UK, with English as their first language and who had at least a Degree or Master's level education in English Language or English Literature, and a 100% Prolific approval rate with at least 200 prior submissions.

[9] https://www.prolific.co

| Task | Model | Open source | Temperature | Score (main subset) | Score (human eval. subset) |
|---|---|---|---|---|---|
| summarisation (ROUGE score) | T0pp | Yes | $0.01^{\dagger}$ | **28.82** | **31.62** |
| | GPT-3 | No | 0 | 24.22 | 27.19 |
| | ChatGPT | No | 0 | 23.76 | 25.72 |
| Simplification (SARI score) | Flan-T5 | Yes | $0.01^{\dagger}$ | **44.98** | **44.61** |
| | InstructGPT | No | 0 | 44.79 | 43.25 |
| | ChatGPT | No | 0 | 37.55 | 35.01 |
| GEC ($F_{0.5}$ score) | OPT-IML | Yes | $0.01^{\dagger}$ | 39.05 | **44.97** |
| | GPT-3 | No | 0 | 38.40 | 41.75 |
| | ChatGPT | No | 0.2 | **39.54** | 37.97 |

Table 1: Automatic evaluation of the best open-source model and two commercial models from OpenAI. Results are shown both on the main subset and the small subset used for human evaluation. $^{\dagger}$Due to the specifics of HuggingFace implementation, a temperature of 0.0 cannot be used, we therefore used a value of 0.01 for such cases.

## 3.3 GPT-4 as a Reviewer

We used GPT-4 as an additional reviewer to assess whether it can be reliably deployed in place of human reviewers. The definition of the evaluation criteria and their assessment scales were included in the GPT-4 prompt together with the input text for each sample.[10] GPT-4 was also asked to annotate the same 100 samples that were shown to human reviewers for each of the three tasks. The full prompts given to GPT-4 for all tasks can also be found in our GitHub repository linked above.

## 4 Results and Discussion

### 4.1 Automatic Evaluation Results

Results are shown in Table 1. In order to allow a comparison between open-source and paid-for models' performance, for each task, we report the best open-source model and two commercial models from OpenAI.[11] For text summarisation, **T0pp** significantly outperformed **GPT-3** and **ChatGPT** (with $p < 0.001$). For text simplification, **Flan-T5** and **InstructGPT** yield the best results, significantly outperforming **ChatGPT** ($p < 0.001$). For GEC, **ChatGPT** and **OPT-IML** perform best with a very similar distribution, significantly outperforming **GPT-3** ($p < 0.001$).

We also observed that for each task, the same prompt seemed to perform best for all models and temperature settings, with only one exception, suggesting that the quality of prompts is almost model-invariant. See Appendix D for more details.

### 4.2 Human and GPT-4 Evaluation Results

Human reviewers and **GPT-4** were shown 4 outputs per sample: the outputs from the models in Table 1 and the gold standard, and were asked to score each model's output on the metrics and scales described in section 3.2. We then converted their scores to rankings for each model and each reviewer from best (1) to worst (4) and took the average.[12] The rankings from human evaluation and **GPT-4** evaluation (in brackets) are shown in Table 2, alongside the interval Krippendorff $\alpha$ coefficient (Krippendorff, 2011) to express inter-annotator agreement. The raw scores and a more detailed set of Krippendorff $\alpha$ coefficients based on individual annotator pairs are shown in Appendix E. As it can be clearly seen, there is generally very good inter-annotator agreement, with an average Krippendorff $\alpha$ of 0.88 across all metrics, with the lowest being 0.62.

On text summarisation, most reviewers scored **ChatGPT** as the best for *Relevance* and *Fluency*, and all reviewers scored **ChatGPT** as best model for *Coherence* and *Consistency*, while **ChatGPT** had a worse ROUGE score compared to other models when using automatic evaluation (see Table 1). Interestingly, all human reviewers scored the gold reference summaries as the worst on all metrics. This reveals the poor quality of reference summaries when compared to most models' outputs, and therefore reference-based automatic metrics could produce unreliable results. It is therefore not surprising that **ChatGPT** outputs were ranked the worst by automatic metrics in text summarisation and simplification, but the best when using human evaluators.

---

[10]Occasionally GPT-4 returned a score of 4.5, and we converted 4.5 to 4 for evaluation purposes (6 out of 3,000 cases).

[11]More detailed results are in Appendix D.

[12]This is to remove subjectivity and individual differences as human reviewers and GPT-4 might employ different marking criteria.

| | Average human annotator rankings (GPT-4 rankings in brackets) | | | |
|---|---|---|---|---|
| Summarisation | RELEVANCE $(\alpha_1^\dagger = 0.88, \alpha_2^\dagger = 0.81)$ | FLUENCY $(\alpha_1 = 0.88, \alpha_2 = 0.82)$ | COHERENCE $(\alpha_1 = 1.00, \alpha_2 = 0.91)$ | CONSISTENCY $(\alpha_1 = 0.97, \alpha_2 = 0.86)$ |
| Gold reference | 4.00 (3.00) | 4.00 (3.00) | 4.00 (3.00) | 4.00 (3.00) |
| T0pp | 3.00 (4.00) | 3.00 (4.00) | 3.00 (4.00) | 2.83 (4.00) |
| GPT-3 | 1.67 (2.00) | 1.67 (**1.50**) | 2.00 (2.00) | 2.17 (2.00) |
| ChatGPT | **1.33 (1.00)** | **1.33** (1.50) | **1.00 (1.00)** | **1.00 (1.00)** |
| Simplification | SEMANTICS $(\alpha_1 = 1.00, \alpha_2 = 0.72)$ | FLUENCY $(\alpha_1 = 1.00, \alpha_2 = 0.50)$ | SIMPLICITY $(\alpha_1 = 0.63, \alpha_2 = 0.63)$ | |
| Gold reference | 4.00 (4.00) | 4.00 (**1.50**) | 3.33 (2.00) | |
| Flan-T5 | **1.00** (2.00) | 3.00 (3.00) | 3.33 (4.00) | |
| InstructGPT | 2.00 (3.00) | 2.00 (4.00) | 2.33 (3.00) | |
| ChatGPT | 3.00 (**1.00**) | **1.00** (1.50) | **1.00 (1.00)** | |
| GEC | SEMANTICS $(\alpha_1 = 0.88, \alpha_2 = 0.34)$ | GRAMMATICALITY $(\alpha_1 = 1.00, \alpha_2 = 0.83)$ | OVER-CORRECTION $(\alpha_1 = 0.62, \alpha_2 = 0.58)$ | |
| Gold reference | 3.33 (2.00) | 3.00 (2.50) | 2.50 (**1.00**) | |
| OPT-IML | **1.00** (4.00) | 4.00 (4.00) | **1.00** (2.00) | |
| GPT-3 | 2.00 (**1.00**) | 2.00 (**1.00**) | 3.00 (3.00) | |
| ChatGPT | 3.67 (3.00) | **1.00** (2.50) | 3.50 (4.00) | |

Table 2: Average human evaluation rankings per model, task and metrics, where 1.00 means best model and 4.00 means worst model. GPT-4 rankings in brackets. When two models were ranked the same, results are shown as average between lower and upper bound (e.g. two best models are shown as 1.50 each). $^\dagger \alpha_1$ represents the interval Krippendorff $\alpha$ coefficient based on the 3 human annotators rankings, while $\alpha_2$ includes GPT-4 rankings.

For text simplification, **ChatGPT** was rated the best model by all reviewers for *Fluency* and *Simplicity*, while it was rated poorly for *Semantics*, with the best model being **Flan-T5**. We observed that this was due to **Flan-T5** returning a lot of outputs which were identical to the inputs, therefore the semantics was obviously fully preserved, but without any inherent text simplification. The gold standard was scored as worst according to all reviewers.

We had substantially different results for GEC, where **ChatGPT** was rated the best model by human reviewers for *Grammaticality* (meaning all or most errors were fixed) but was rated as worst or second worst model for *Semantics* and *Over-correction*, for which the best model was **OPT-IML**. This underlines how **ChatGPT** tends to over-correct, and in doing so might add information to the sentence which were not originally present, which is consistent with recent findings (Fang et al., 2023; Wu et al., 2023). The gold reference was scored mostly as second worst on most metrics and by most reviewers.

For both text summarisation and simplification, **GPT-4** used as a reviewer produced surprisingly good results which correlate well, albeit not perfectly, with human reviewers. We observed a stronger disagreement between human reviewers and **GPT-4** in GEC. It is also worth noting that we did not observe the systematic positional bias when using **GPT-4** as a reviewer as reported by Wang et al. (2023). However, we postulate that averaging the scores across the samples and using rankings instead of absolute scores helped to dampen this effect. If we include **GPT-4** evaluation, the average Krippendorff $\alpha$ is 0.70 across all metrics, with the lowest being 0.34.

## 5 Conclusion

Model evaluation is a topic which is attracting increasing interest from the community. Liang et al. (2022) have recently published an extensive evaluation report on LLMs, however they mostly focused on automatic evaluation. Prompted by the recent advances in generative capabilities of the latest LLMs, we conducted this study to explore the drift between human judgement and automatic, reference-based evaluation of zero-shot model performance. We also explored model-to-model evaluation with GPT-4. The study was conducted using large, open-source datasets often acting as benchmarks for their respective tasks.

Our work reveals a systematic misalignment between reference-based automatic metrics and human evaluation on a range of generative tasks, highlighting the inadequacy of the gold reference in the public NLP benchmarks. It is not clear whether this misalignment is purely due to the limitations

of automatic metrics, or whether poor reference quality makes using any reference-based comparative metrics unreliable. Despite ChatGPT being rated one of the best models on most metrics by human reviewers, the best open-source LLMs also consistently outperformed the reference outputs. We also explored the potential of GPT-4 to act as a reviewer and found it has strong correlation with human judgement for summarisation and simplification tasks, and moderate correlation for GEC.

Future work will look at improving the quality of prompts, providing few-shot in-context learning (Brown et al., 2020), or exploring the potential of chain-of-thought prompting (Wei et al., 2022) in improving models' outputs. Given the misalignment mentioned above, extending human evaluation to larger datasets and to a wider range of model settings will also be of particular future interest, so as to minimise the bias introduced when using automatic metrics to select a subset for human evaluation. Finally, introducing multiple automatic evaluation metrics (e.g. reference-less) for each task might help deepen our understanding of the relation between such metrics and human judgement.

## Limitations

This paper suffers from the following limitations:

- A limited amount of prompt tuning and prompt space investigation was carried out. Between 2 and 5 different prompts per task were tried, therefore a more focused study on prompt engineering could potentially bring significant improvements, however this is a stand-alone exploration topic, which we leave for future work.

- We did not perform any in-context learning or chain-of-thought prompting, which have been shown to significantly improve the performance of generative models. As such, there may be margin for improving the quality of models' outputs, while the quality of gold references will remain unchanged until new datasets become available.

- We used automatic metrics (SARI, ROUGE and $F_{0.5}$) to determine the best combination of settings (model, prompt, temperature) for each task. However, since this study revealed poor correlation between human judgement and such metrics, we cannot exclude that the

settings we chose for human evaluation were not the most appropriate, which means the study may have suffered from some bias indirectly introduced by using automatic metrics for selection of outputs for the human evaluation study. This is further aggravated by traditional open source datasets only presenting one gold reference output per sample when multiple equally valid outputs could exist, leading to unreliable scores; for example, two summaries of the same story can be both very good but contain few common bi-grams, leading to a poor ROUGE score when doing automatic evaluation.

- Given the wide variety of the text corpora on which most of the models we used were pretrained on, it is very likely that at least some of the models may have been trained on some of the open-source datasets we used to evaluate them. While it is difficult to mitigate for this (for example OpenAI did not publish a list of datasets used to train their models), our results might have been affected by this, and using new unreleased datasets would have been preferable to reduce this bias. However, this was not possible due to the highly expensive and time consuming nature of the task of creating high quality large datasets from scratch, which is a well known issue across the research community.

- While we did not use the same model for both inference and evaluation, we used GPT-4 for evaluation of all models, including the outputs from ChatGPT. Considering they belong to the same family of OpenAI models, GPT-4 might have a bias for rating ChatGPT's outputs higher than other models. However, our results were not able to validate or refute this, as human reviewers also rated ChatGPT outputs as the best across most metrics.

- Due to time and budgetary constraints, we were only able to hire 3 reviewers (not including GPT-4), and asked reviewers to annotate 100 samples per dataset, which is a small proportion of each dataset. Due to the small number of reviewers and reviewed samples, the noise-to-signal ratio may affect the strength and generalisability of our findings. Furthermore, using human evaluation as gold standard is also prone to introducing bias. How-

ever, we found that in most cases all annotators agreed that the gold standard was worse than the best models' outputs, so we do believe this is a valid conclusion, given how consistent it was across different tasks and annotators.

## Ethics Statement

Our work makes use of LLMs, and there are known concerns associated with such models (Bender et al., 2021), including data bias, toxicity of training content or outputs, their environmental impact, the lack of explainability for their outputs, and the potential to replace human workers with resulting job losses. We did not perform any fine-tuning as part of this project, and only used open-source datasets. Some of the OpenAI's models we used are not open-source, and their overall impact on society is only starting to become apparent. Overall we believe this research does not increase the risk of harm caused by these models or datasets as we only explored their limitations and performance. We employed 3 human annotators through the Prolific platform for a 16-hour study. Reviewers were paid £13.20 per hour, not including Prolific's fees.[13] We did not collect any personal information beyond demographic data provided by Prolific, including age, profession, gender amongst others. While Prolific does provide such data, we did not use them as screening criteria, and only adopted the screening criteria mentioned in section 3.2. All annotators were provided with a detailed description of the study before committing to take part.

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

## A  Newsela Dataset Processing

We observed that the ACL 2020 version (Jiang et al., 2020) of the Newsela dataset (Xu et al., 2015) contains a number of samples where either the source (input) or the destination (reference) were duplicated. In such cases, based on our observations, it was appropriate to merge them into a single sample. If the source was a duplicate but the destination wasn't, we kept the source without duplication, and created the destination by merging the two original destination samples, in the order in which they appear in the dataset. Likewise if the destination was a duplicate but the source wasn't. See example below

- Original dataset, sample 1

  - Source: *Ron Bee , a professor at San Diego State University , is worried that so few Americans serve in the military .*
  - Destination: *Ron Bee is a professor in California , and he is worried .*

- Original dataset, sample 2

  - Source: *Ron Bee , a professor at San Diego State University , is worried that so few Americans serve in the military .*
  - Destination: *Very few people join the military now .*

- Our merged sample

  - Source: *Ron Bee , a professor at San Diego State University , is worried that so few Americans serve in the military .*
  - Destination: *Ron Bee is a professor in California , and he is worried . Very few people join the military now .*

## B  Implementation Details

Due to time and budgetary constraints, the full scale experiments were performed using the most promising settings after a preliminary study conducted on a subset of each dataset (which consists of 100 samples) on a much broader range of settings. We experimented with a range of prompts and temperature values to better explore the capabilities of each model. The final settings are task dependent; for example, we empirically observed that lower temperature values always gave the best outcomes for text summarisation and simplification, whereas for GEC it was beneficial to use higher values for some models.

### B.1  Prompt Engineering

The following prompts were used, where \n indicates a newline and [...] indicates the input sample; for each of the three tasks, we report the best prompt, i.e. the prompt whose output was used for our evaluation work, at the top (prompt (a)). The same prompt yielded best results regardless of model and temperature, with extremely limited exceptions.

1. Text summarisation

   (a) `Summarize the following text. [...] \n The summary is:`
   (b) `[...] \n Summarize the text above.`

   (c) `Summarize  the  following  text.` `[...] \n The very short summary` `is:`

   (d) `This is the main story:  [...]` `\n The summarized version of the` `story is:`

2. Text simplification

   (a) `Simplify  the  following  text.` `[...] \n The simplified version` `is:`

   (b) `This is the main story: [...]\n` `The simplified version of the` `story is:`

   (c) `Simplify  the  following  text.` `[...]`

   (d) `Explain this to a 5 year old.` `[...]`

   (e) `Explain this to a 5 year old.` `[...]  \n The explanation to a` `5 year old could be:`

3. Grammatical Error Correction

   (a) `Reply with a corrected version` `of the input sentence with all` `grammatical and spelling errors` `fixed.  If there are no errors,` `reply with a copy of the original` `sentence.  \n\n Input sentence:` `[...] \n Corrected sentence:`

   (b) `Correct the following to standard` `English: \n\n Sentence: [...] \n` `Correction:`

When using GPT-4 as a reviewer, we prompted GPT-4 to output the text following strict json format rules so its output could be processed at scale programmatically. When it failed to do so, we re-run the evaluation on that specific sample until the output was in the desired format, which happened mostly at the first attempt and occasionally after 2-3 attempts as GPT-4 output is non-deterministic.

## B.2 Hyperparameter Tuning

We experimented with the following temperature values: 0.0 (we used 0.01 for HuggingFace models due to implementation requirements), 0.2, 0.5, 0.7. We observed that for text simplification and summarisation, the lowest value always yielded the best results, whereas for GEC, some combinations of models and prompts yielded better results for

temperatures of 0.2 or 0.5, despite the best overall combination being at a temperature of 0.0 even for GEC. For all other hyper-parameters, we used the default settings for each model without modifications.

## B.3 Tokenization and Truncation

While the Newsela and BEA-2019 dataset samples are all below 512 tokens,[14] the samples from CNN / DailyMail have a broader distribution, with 80.6% exceeding 482 tokens and 9.8% exceeding 1506 tokens. Different models and implementations have different maximum sequence lengths. Furthermore, while OpenAI models count the total number of input and output tokens towards their maximum sequence length, HuggingFace models have two separate limits for input and output tokens respectively. In order to facilitate the inference process, we used the following heuristics to tailor different design decisions to each model to try to maximise performance:

- For GPT-3, which accepts up to 4000 combined input and output tokens, we did not perform any truncation, as the longest sample had 2,571 tokens.

- For InstructGPT, which accepts up to 2049 combined input and output tokens, we truncated the input after 1506 tokens. This leaves 512 tokens for the generated output, as well as a further 31 tokens for the prompt (it is imperative not to truncate the portion of the prompt at the end of the input)

- For HuggingFace models accepting inputs up to 512 tokens (excluding the output), we truncated at 482 tokens to leave space for the prompt; for HuggingFace models accepting inputs up to 2048 tokens we truncated at 2018 tokens.

## C  Human Evaluation Criteria for GEC

The criteria and their definitions and assessment scales given to reviewers for the GEC task are reported below.

- *Semantics*. This assesses whether the meaning of the text is preserved following the GEC. Semantic preservation is assessed on a 5-point

---

[14]We used `text-davinci-003` as tokenizer with the `tiktoken` python library, however we observed negligible differences when using HuggingFace tokenizers.

Likert scale from 1 (Meaning Not Preserved) to 5 (Meaning fully preserved). NOTE: You should penalise corrections which change the meaning unnecessarily. For example, the sentence "I wentt at Rome for my birthday" should be corrected to "I went to Rome for my birthday". A correction such as "I went to Rome for my anniversary" should be penalised in this category as they introduce unnecessary changes to the meaning.

- *Grammaticality*. This assesses the quality of the correction and answers the question "How many errors are left in the corrected sentence?". Please provide a count of the remaining errors, regardless of whether they were present in the source or they were newly introduced errors in the supposed corrected version. The three options are "0", "1", "2 or more".

- *Over-correction*. Since there can be multiple ways to correct a sentence, this assesses whether the correction is unnecessarily verbose or makes unnecessary syntax changes. The best correction should be done with the minimum number of edits. For example, if the sentence "I wentt at Rome for my birthday" is corrected to "I decided to go to Rome for my birthday" this should be penalised under this category because it contains unnecessary syntax changes, even though the final sentence is grammatically correct. This metric answers the question: Is the system over-correcting or making unnecessary syntax changes? The answers should be "No", "Minor over-correction", "Moderate over-correction" or "Substantial over-correction".

Note that a correction which results in a change of meaning will most likely also be an over-correction. Therefore we expect that if a correction is given a poor score in the Semantics category, it will also receive a poor score in the Over-correction category, and as such there may be some overlap between these two metrics. However, the reverse is not necessarily true, as you could easily have an over-correction without a change of meaning. For example, correcting a sentence from "I wentt at Rome for my birthday" to "I decided to go to Rome for my birthday" doesn't significantly affect the

meaning of the sentence, but it nonetheless represents a clear case of over-correction as "wentt at" should have been corrected to "went to" instead of "decided to go to". As such we felt there would be value in keeping these two metrics separate.

# D    Detailed Automatic Evaluation Results

Table 3 shows the average results of the experiments we run on the summarisation dataset, for each model, temperature and prompt. Refer to Appendix B.1 for prompt details. Table 4 shows the average results of the experiments we run on the simplification dataset. Table 5 shows the average results of the experiments we run on the GEC dataset.

| Model | Temp. | Summ. Prompt | ROUGE score |
|---|---|---|---|
| ChatGPT | 0 | 1a | 23.76 |
| GPT-3 | 0 | 1a | 24.22 |
| | | 1b | 23.28 |
| | 0.7 | 1a | 23.24 |
| | | 1b | 22.46 |
| InstructGPT | 0 | 1a | 20.04 |
| | | 1b | 18.60 |
| | 0.7 | 1a | 19.60 |
| | | 1b | 19.04 |
| T0pp | 0.01 | 1a | 28.82 |
| | | 1b | 28.80 |
| | 0.7 | 1a | 26.31 |
| | | 1b | 26.19 |
| Flan-UL2 | 0.01 | 1a | 23.77 |
| | | 1b | 18.61 |
| | 0.7 | 1a | 21.83 |
| | | 1b | 17.25 |

Table 3: Detailed automatic evaluation results on the text summarisation task.

| Model | Temp. | Simp. Prompt | SARI score |
|---|---|---|---|
| ChatGPT | 0 | 2a | 37.55 |
| GPT-3 | 0 | 2a | 36.03 |
| | | 2b | 36.60 |
| | 0.5 | 2a | 35.81 |
| | | 2b | 36.47 |
| | 0.7 | 2a | 35.73 |
| | | 2b | 36.42 |
| InstructGPT | 0 | 2a | 44.79 |
| | | 2b | 43.44 |
| | 0.5 | 2a | 43.51 |
| | | 2b | 42.53 |
| | 0.7 | 2a | 42.63 |
| | | 2b | 40.84 |
| OPT-IML | 0.01 | 2a | 41.33 |
| | | 2b | 38.21 |
| | 0.5 | 2a | 40.79 |
| | | 2b | 37.99 |
| | 0.7 | 2a | 40.45 |
| | | 2b | 37.52 |
| Flan-T5 | 0.01 | 2a | 44.98 |
| | | 2b | 38.11 |
| | 0.5 | 2a | 43.98 |
| | | 2b | 36.97 |
| | 0.7 | 2a | 42.79 |
| | | 2b | 35.59 |
| T0pp | 0.01 | 2a | 43.87 |
| | | 2b | 40.63 |
| | 0.5 | 2a | 42.48 |
| | | 2b | 39.59 |
| | 0.7 | 2a | 41.00 |
| | | 2b | 38.94 |
| Flan-UL2 | 0.01 | 2a | 43.43 |
| | | 2b | 35.75 |
| | 0.5 | 2a | 42.11 |
| | | 2b | 34.87 |
| | 0.7 | 2a | 41.00 |
| | | 2b | 33.94 |

Table 4: Detailed automatic evaluation results on the text simplification task.

| Model | Temp. | GEC Prompt | $F_{0.5}$ score |
|---|---|---|---|
| ChatGPT | 0 | 3a | 39.48 |
| | | 3b | 22.47 |
| | 0.2 | 3a | 39.54 |
| | | 3b | 22.38 |
| GPT-3 | 0 | 3a | 38.40 |
| | | 3b | 33.39 |
| | 0.2 | 3a | 38.31 |
| | | 3b | 33.28 |
| | 0.5 | 3a | 37.86 |
| | | 3b | 32.22 |
| InstructGPT | 0 | 3a | 40.44 |
| | | 3b | 38.92 |
| | 0.2 | 3a | 40.00 |
| | | 3b | 37.56 |
| | 0.5 | 3a | 36.77 |
| | | 3b | 31.52 |
| OPT-IML | 0.01 | 3a | 39.05 |
| | | 3b | 36.04 |
| | 0.3 | 3a | 38.36 |
| | | 3b | 35.52 |
| | 0.6 | 3a | 35.84 |
| | | 3b | 33.70 |
| | 0.9 | 3a | 29.89 |
| | | 3b | 27.38 |
| Flan-T5 | 0.01 | 3a | 19.17 |
| | | 3b | 12.84 |
| | 0.3 | 3a | 20.64 |
| | | 3b | 13.83 |
| | 0.6 | 3a | 22.33 |
| | | 3b | 15.54 |
| | 0.9 | 3a | 20.62 |
| | | 3b | 15.40 |

Table 5: Detailed automatic evaluation results on the GEC task.

## E  Detailed Human and GPT-4 Evaluation Results

Table 6 shows the average scores and standard deviation for each human reviewer and GPT-4 for each task, metric and model across the three 100-sample subsets. Table 7 shows the same data by ranking, instead of absolute scores. It also shows the interval Krippendorff $\alpha$ coefficient expressing inter-annotator agreement for all annotator pairs, including each annotator and GPT-4.

| | Metric | Model | Avg. annot. 1 | Std. dev. annot. 1 | Avg. annot. 2 | Std. dev. annot. 2 | Avg. annot. 3 | Std. dev. annot. 3 | Avg. GPT-4 | Std. dev. GPT-4 |
|---|---|---|---|---|---|---|---|---|---|---|
| **Text summarisation** | RELEVANCE | Gold reference | 3.06 | 0.97 | 3.57 | 1.11 | 3.89 | 1.18 | 4.46 | 0.61 |
| | | T0pp | 3.44 | 0.98 | 3.89 | 0.87 | 4.08 | 1.02 | 3.94 | 0.54 |
| | | GPT-3 | 4.61 | 0.68 | 4.47 | 0.68 | 4.67 | 0.60 | 4.68 | 0.47 |
| | | ChatGPT | 4.91 | 0.38 | 4.79 | 0.43 | 4.59 | 0.63 | 4.84 | 0.37 |
| | FLUENCY | Gold reference | 2.24 | 1.14 | 3.40 | 0.98 | 1.87 | 0.69 | 4.98 | 0.14 |
| | | T0pp | 3.44 | 1.20 | 4.66 | 0.78 | 4.30 | 1.28 | 4.93 | 0.26 |
| | | GPT-3 | 4.62 | 0.70 | 4.92 | 0.44 | 4.84 | 0.48 | 5.00 | 0.00 |
| | | ChatGPT | 4.85 | 0.52 | 4.96 | 0.24 | 4.82 | 0.55 | 5.00 | 0.00 |
| | COHERENCE | Gold reference | 2.87 | 1.43 | 3.64 | 1.32 | 3.51 | 1.51 | 4.50 | 0.52 |
| | | T0pp | 3.90 | 1.18 | 4.60 | 0.88 | 4.20 | 1.18 | 3.72 | 0.60 |
| | | GPT-3 | 4.80 | 0.53 | 4.79 | 0.43 | 4.78 | 0.67 | 4.52 | 0.52 |
| | | ChatGPT | 4.90 | 0.41 | 4.87 | 0.36 | 4.87 | 0.48 | 4.81 | 0.39 |
| | CONSISTENCY | Gold reference | 4.48 | 0.87 | 3.73 | 1.15 | 4.05 | 1.37 | 4.74 | 0.46 |
| | | T0pp | 4.90 | 0.41 | 4.38 | 0.85 | 4.83 | 0.69 | 4.21 | 0.57 |
| | | GPT-3 | 4.96 | 0.20 | 4.68 | 0.61 | 4.83 | 0.65 | 4.76 | 0.43 |
| | | ChatGPT | 4.98 | 0.20 | 4.76 | 0.45 | 4.89 | 0.44 | 4.88 | 0.32 |
| **Text Simplification** | SEMANTICS | Gold reference | 2.78 | 1.31 | 3.39 | 1.31 | 1.91 | 1.43 | 4.03 | 1.01 |
| | | Flat-T5 | 4.97 | 0.17 | 4.86 | 0.53 | 4.83 | 0.66 | 4.55 | 0.95 |
| | | InstructGPT | 4.78 | 0.72 | 4.83 | 0.60 | 4.67 | 0.94 | 4.48 | 0.96 |
| | | ChatGPT | 4.42 | 0.83 | 4.58 | 0.85 | 3.98 | 1.30 | 4.66 | 0.57 |
| | FLUENCY | Gold reference | 3.70 | 0.69 | 3.67 | 1.33 | 2.10 | 0.70 | 5.00 | 0.00 |
| | | Flat-T5 | 4.64 | 0.57 | 4.64 | 0.79 | 3.80 | 1.25 | 4.99 | 0.10 |
| | | InstructGPT | 4.76 | 0.74 | 4.80 | 0.68 | 4.33 | 0.87 | 4.95 | 0.41 |
| | | ChatGPT | 4.77 | 0.65 | 4.93 | 0.32 | 4.47 | 0.92 | 5.00 | 0.00 |
| | SIMPLICITY | Gold reference | 3.10 | 1.05 | 3.81 | 1.17 | 3.35 | 1.28 | 3.91 | 0.60 |
| | | Flat-T5 | 3.39 | 0.63 | 3.89 | 0.79 | 3.09 | 0.35 | 3.19 | 1.21 |
| | | InstructGPT | 3.41 | 0.72 | 4.04 | 0.79 | 3.14 | 0.58 | 3.27 | 1.16 |
| | | ChatGPT | 4.52 | 0.70 | 4.69 | 0.70 | 4.15 | 1.06 | 4.25 | 0.62 |
| **Grammatical Error Correction** | SEMANTICS | Gold reference | 4.91 | 0.38 | 4.74 | 0.59 | 4.77 | 0.66 | 4.86 | 0.51 |
| | | OPT-IML | 5.00 | 0.00 | 4.94 | 0.37 | 4.97 | 0.22 | 4.37 | 0.73 |
| | | GPT-3 | 4.95 | 0.26 | 4.75 | 0.65 | 4.88 | 0.41 | 4.88 | 0.35 |
| | | ChatGPT | 4.89 | 0.58 | 4.68 | 0.86 | 4.79 | 0.74 | 4.79 | 0.73 |
| | GRAMMATICALITY† | Gold reference | 0.64 | 0.77 | 0.38 | 0.61 | 0.42 | 0.72 | 0.12 | 0.38 |
| | | OPT-IML | 1.00 | 0.76 | 0.58 | 0.67 | 0.83 | 0.85 | 0.62 | 0.58 |
| | | GPT-3 | 0.54 | 0.71 | 0.18 | 0.41 | 0.26 | 0.56 | 0.04 | 0.24 |
| | | ChatGPT | 0.47 | 0.67 | 0.15 | 0.38 | 0.24 | 0.55 | 0.12 | 0.41 |
| | OVERCORRECTION† | Gold reference | 0.05 | 0.30 | 0.45 | 0.75 | 0.40 | 0.79 | 0.00 | 0.00 |
| | | OPT-IML | 0.00 | 0.00 | 0.14 | 0.55 | 0.01 | 0.10 | 0.12 | 0.35 |
| | | GPT-3 | 0.04 | 0.28 | 0.56 | 0.86 | 0.46 | 0.84 | 0.18 | 0.41 |
| | | ChatGPT | 0.06 | 0.34 | 0.61 | 1.01 | 0.40 | 0.82 | 0.19 | 0.52 |

Table 6: Average and standard deviation of the scores given by each human annotator and GPT-4 per model, task and metrics, across all analysed samples (100 per task). †All metrics on a Likert scale 1 (worst) to 5 (best) except Grammaticality on a scale 0 (best) to 3 (worst) and Over-Correction on a scale 0 (best) to 2 (worst).

| | Metric | Model | Avg. annot. 1 (rank) | Avg. annot. 2 (rank) | Avg. annot. 3 (rank) | Avg. GPT-4 (rank) | Kripp. α annot. 1 and 2 | Kripp. α annot. 2 and 3 | Kripp. α annot. 1 and 3 | Kripp. α annot. 1 and GPT-4 | Kripp. α annot. 2 and GPT-4 | Kripp. α annot. 3 and GPT-4 |
|---|---|---|---|---|---|---|---|---|---|---|---|---|
| **Text summarisation** | RELEVANCE | Gold reference | 4 | 4 | 4 | 3 | | | | | | |
| | | T0pp | 3 | 3 | 3 | 4 | 1.00 | 0.83 | 0.83 | 0.83 | 0.83 | 0.65 |
| | | GPT-3 | 2 | 2 | 1 | 2 | | | | | | |
| | | ChatGPT | 1 | 1 | 2 | 1 | | | | | | |
| | FLUENCY | Gold reference | 4 | 4 | 4 | 3 | | | | | | |
| | | T0pp | 3 | 3 | 3 | 4 | 1.00 | 0.83 | 0.83 | 0.77 | 0.77 | 0.77 |
| | | GPT-3 | 2 | 2 | 1 | 1.5 | | | | | | |
| | | ChatGPT | 1 | 1 | 2 | 1.5 | | | | | | |
| | COHERENCE | Gold reference | 4 | 4 | 4 | 3 | | | | | | |
| | | T0pp | 3 | 3 | 3 | 4 | 1.00 | 1.00 | 1.00 | 0.83 | 0.83 | 0.83 |
| | | GPT-3 | 2 | 2 | 2 | 2 | | | | | | |
| | | ChatGPT | 1 | 1 | 1 | 1 | | | | | | |
| | CONSISTENCY | Gold reference | 4 | 4 | 4 | 3 | | | | | | |
| | | T0pp | 3 | 3 | 2.5 | 4 | 1.00 | 0.95 | 0.95 | 0.83 | 0.83 | 0.68 |
| | | GPT-3 | 2 | 2 | 2.5 | 2 | | | | | | |
| | | ChatGPT | 1 | 1 | 1 | 1 | | | | | | |
| **Text Simplification** | SEMANTICS | Gold reference | 4 | 4 | 4 | 4 | | | | | | |
| | | Flan-T5 | 1 | 1 | 1 | 2 | 1.00 | 1.00 | 1.00 | 0.48 | 0.48 | 0.48 |
| | | InstructGPT | 2 | 2 | 2 | 3 | | | | | | |
| | | ChatGPT | 3 | 3 | 3 | 1 | | | | | | |
| | FLUENCY | Gold reference | 4 | 4 | 4 | 1.5 | | | | | | |
| | | Flan-T5 | 3 | 3 | 3 | 3 | 1.00 | 1.00 | 1.00 | 0.03 | 0.03 | 0.03 |
| | | InstructGPT | 2 | 2 | 2 | 4 | | | | | | |
| | | ChatGPT | 1 | 1 | 1 | 1.5 | | | | | | |
| | SIMPLICITY | Gold reference | 4 | 4 | 2 | 2 | | | | | | |
| | | Flan-T5 | 3 | 3 | 4 | 4 | 1.00 | 0.48 | 0.48 | 0.48 | 0.48 | 1.00 |
| | | InstructGPT | 2 | 2 | 3 | 3 | | | | | | |
| | | ChatGPT | 1 | 1 | 1 | 1 | | | | | | |
| **Grammatical Error Correction** | SEMANTICS | Gold reference | 3 | 3 | 4 | 2 | | | | | | |
| | | OPT-IML | 1 | 1 | 1 | 4 | 1.00 | 0.83 | 0.83 | -0.05 | -0.05 | -0.23 |
| | | GPT-3 | 2 | 2 | 2 | 1 | | | | | | |
| | | ChatGPT | 4 | 4 | 3 | 3 | | | | | | |
| | GRAMMATICALITY | Gold reference | 3 | 3 | 3 | 2.5 | | | | | | |
| | | OPT-IML | 4 | 4 | 4 | 4 | 1.00 | 1.00 | 1.00 | 0.68 | 0.68 | 0.68 |
| | | GPT-3 | 2 | 2 | 2 | 1 | | | | | | |
| | | ChatGPT | 1 | 1 | 1 | 2.5 | | | | | | |
| | OVERCORRECTION | Gold reference | 3 | 2 | 2.5 | 1 | | | | | | |
| | | OPT-IML | 1 | 1 | 1 | 2 | 0.83 | 0.68 | 0.40 | 0.48 | 0.83 | 0.40 |
| | | GPT-3 | 2 | 3 | 4 | 3 | | | | | | |
| | | ChatGPT | 4 | 4 | 2.5 | 4 | | | | | | |

Table 7: Average rankings given by each human annotator and GPT-4 per model, task and metrics, where 1 means best model and 4 means worst model. Results are based on averaged scores across all analysed samples (100 per task). The last 6 columns represent the interval Krippendorff $\alpha$ coefficients expressing inter-annotator agreement for all annotator pairs, including each annotator and GPT-4. The Krippendorff $\alpha$ coefficients are shown by task and metric, but aggregated across all models. Krippendorff $\alpha$ can also be computed with more than two annotators. For inter-annotator agreements based on all human annotators combined, with and without GPT-4 annotations, see Table 2.