# OpenReview forum: "Evaluation Metrics in the Era of GPT-4: Reliably Evaluating Large Language Models on Sequence to Sequence Tasks"
_EMNLP/2023/Conference — EMNLP 2023 Main_

### Official Review · Reviewer_JKmf · 2023-08-02

**Soundness:** 4

**Excitement:**

4: Strong: This paper deepens the understanding of some phenomenon or lowers the barriers to an existing research direction.

**Paper Topic And Main Contributions:**

The paper presents a hybrid evaluation of recent LLMs on three NLP tasks: text simplification, text summarization, and grammatical error correction. They ask human evaluators to judge the output quality across task-specific and task-agnostic dimension criteria and find a discrepancy between the rankings of automatic metrics and human evaluation and argue that automatic metrics are not reliable even for system development. Further, they show that rankings induced by ChatGPT as an evaluator better approximates human judgments.

**Questions For The Authors:**

1. How were the outputs presented to the prolific workers? Could you also include detailed instructions for the same?
2. How are the rankings induced by automatic metrics from Table 1 different from ChatGPT? I would suggest breaking down Table 2 in two to highlight: a) rankings induced by human eval and b) the ability of automatic metrics to recover system ranking.
3. How well do automatic metrics correlate with segment-level rankings?

**Reasons To Accept:**

1. The paper presents an important data point for the re-consideration of popular benchmarks as the output quality of references is judged lower than model-generated outputs.
2. The paper presents a systematic evaluation of multiple LLMs on three tasks that can be useful for development in human and automatic evaluation.


**Reasons To Reject:**

1. The scale of the evaluation (100 examples) is small to make strong claims (e.g. l76: we prove that...). It might be useful to note the scale limitation in the paper when mentioning the findings.
2. While authors note that selecting models for human eval via automatic metrics could have biased the evaluation in limitations, it is equally important to acknowledge the limitations of the human evaluation as a gold standard itself. It would be useful to provide some discussion in this direction either in the main text or the limitation section.

**Reproducibility:**

4: Could mostly reproduce the results, but there may be some variation because of sample variance or minor variations in their interpretation of the protocol or method.

**Reviewer Confidence:**

4: Quite sure. I tried to check the important points carefully. It's unlikely, though conceivable, that I missed something that should affect my ratings.

---

> ### Author Rebuttal · Authors · 2023-08-29
>
> We thank Reviewer 3 for the positive comments. Please see below our responses to their main concerns and questions.
>
> - **Reason to Reject 1**: We agree that 100 samples for human evaluation is not a large subset to make very strong claims. Unfortunately human evaluation experiments take a lot of time for the reviewers to carry out, and are also considerably expensive for the researchers. As such we decided to limit the amount of samples used for this study. We are more than happy to add a note to the Limitations section in the camera-ready version regarding potential limitations of the strength of the findings given the sample size. However, we found that in most cases all annotators agreed that the gold standard was worse than some models’ outputs, so we do believe this is a valid conclusion, given how consistent it was across different tasks and annotators.
> - **Reason to Reject 2**: We fully take on-board the point about the limitations of human evaluation. While human evaluation done by experts is the gold-standard in this study, there can be a degree of subjectivity in evaluating language model outputs, and therefore using a much larger number of reviewers (for example 50 instead of 3) would make the findings much stronger and reduce potential noise. This would require substantially more financial expenditures on the human evaluation task. We will happily add a note about this in the camera-ready version to emphasize this aspect.
>
>
> - **Question 1**: The evaluation survey for each of the three tasks had a landing page which contained detailed information about the task and evaluation criteria. Once the reviewer has familiarized themselves they click on a “Proceed” button which takes them to the first sample. Each sample was on a different webpage to avoid confusion, and the users were able to move between samples using browser buttons. On each page, the main input text was shown at the top. The user was then shown four different outputs below each input, and below each output they were asked to rate the output above using the task-specific criteria outlined in the paper. They were able to see all four outputs for the same input on the same webpage, making it easier to give scores which were correct relative to each other. The detailed instructions given are very long and attached at the end of this rebuttal.
> - **Question 2**: There are two issues with using automatic metrics to recover rankings, in particular 1) automatic metrics cannot be used to rank the gold standard references, as they require such references for their ranking, and as a result the gold standard would always be ranked as the best by reference-based automatic metrics, whereas using human evaluation allows us to include the gold standard in the outputs to rank, 2) the automatic metrics we used only return a single score for each task, whereas for human evaluation we used multiple task-specific criteria (such as relevance, fluency, coherence etc.). We could potentially average these criteria and get a single human ranking score for each model and compare that to the rankings generated using automatic metrics, however given we would not be able to include the gold standard in the comparison, we chose not to do this.
> - **Question 3**: Overall automatic metrics correlate poorly with segment-level rankings. First of all, the gold-reference used by automatic metrics ranks worst or second worst on almost all metrics for all tasks, as shown in Table 2. If we exclude the gold-reference from the comparison, we still note that, for example, ChatGPT is rated as the worst model using automatic metrics for both summarization and simplification (Table 1), whereas it’s one of the best models according to human evaluators.
>
> Below are the instructions given to human reviewers and GPT4 by task (as requested in question 1). The human reviewers instructions contain html tags as they were delivered as a UI.
>
> - HUMAN REVIEWERS INSTRUCTIONS FOR TEXT SIMPLIFICATION
>
> ```html
> <big>Human evaluation of text simplification</big> <div style="height: 15px;" />IMPORTANT INFORMATION: We are conducting this research on the quality of text simplification, summarization, and grammatical error correction. We will start from text simplification. You will be asked to read some short texts and their simplified versions and rate them according to some criteria which are specified below. Some of the content you will see is human generated, other is AI generated, but you will not be told which is which as this is a blind research. As the interface is not mobile / tablet friendly, we ask that you complete this study using a laptop or desktop. This research is divided into 3 parts taking about 4, 8 and 4 hours respectively (around 16 hours in total). You should only start working on this research if you can at least commit for the first part (4 hours). To make things simpler, we are splitting these 4 hours into 4 separate studies taking about one hour each, which you can take on separate days. All these 4 one-hour studies should be completed by the end of Thursday 4th May. This is study part 1 of 4. A link to the following 3 studies will be sent once you have accepted this commitment. Base pay is £12 per hour. Once you've completed the first 4 one-hour studies, we will review the work, and if of a satisfactory quality, we will award a 10% discretionary bonus and invite you to take part to the two remaining parts of the study which will require a longer time commitment. You can decide not to continue with the research after you have completed the first 4 one-hour studies. Work will be reviewed as it comes through during UK working hours (excluding weekends and bank holidays). If at any point you have any questions or we have anything we'd like to communicate, feel free to message us through the Prolific messaging service, we will also use it if we need to get in touch, and potentially to discuss continuation of your work into the following studies. <div style="height: 15px;" /> <ul>  In the study you're about to start, which we estimate should take one hour, you will be shown 25 pieces of text taken mostly from older news articles. Alongside each of them, you will be shown 4 different possible simplified versions of the text. The purpose of text simplification can be varied, but include things such as making the text easier to read, more accessible to language learners, or explaining a medical or legal text in plain speak. We ask that you carefully read the original text and rank each of the 4 simplifications according to the following metrics, which are defined below. <div style="height: 10px;" /> A simplification box would look as follows (you will have 4 of these per source text) <head><style>* {margin: 0; padding: 0;} .imgbox {display: grid; height: 100%;} .center-fit { max-width: 100%; max-height: 100vh; margin: auto;}</style></head><body><div class="imgbox"><img class="center-fit" src='surveyflow/screenshot.png'></div></body>  <strong>NOTE: please take a screeshot of this page for your future reference as you may not be able to access it again once you start the survey.</strong><div style="height: 15px;" /> <ul>           <li>                <p><strong>Semantics (or adequacy)</strong> </p> This assesses whether the meaning of the text is preserved further to the simplification. Semantic preservation is assessed on a 5-point Likert scale from 1 (Meaning Not Preserved) to 5 (Meaning fully preserved). NOTE: You should penalize simplified text which is not self contained. For example, if the sentence <em>"Scientists have no idea how eels protect their own brain and muscles from being damaged by electricity"</em> is simplified as <em>"Scientists have no idea how they stay safe"</em>, this should be rated around 2 out of 5 as it is not possible by just reading the simplified version to understand what "they" is referring to. Like wise if the simplified text contains hallucinations (facts not contained in the source) it should be penalised. For example if <em>"He yelled"</em> is simplified as <em>"His father Russell shouted"</em> this should be rated around 2 under this category. The omission of some unimportant details from the source into the simplified version is acceptable at your discretion, if the text meaning is preserved. </li> <div style="height: 15px;" /> <li> <p><strong>Grammaticality (or fluency)</strong></p>This assesses whether the simplified text remains grammatical and understandable. Sentences should have no formatting problems, capitalization errors or obviously ungrammatical sentences (e.g., fragments, missing components) that make the text difficult to read. Grammaticality / Fluency is assessed on a 5-point Likert scale from 1 (Not Fluent) to 5 (Super Fluent)</li>  <div style="height: 15px;" /> <li> <p><strong>Simplicity</strong></p> This assesses whether the simplified text, considered in its entirety, is simpler than the source text. Simplicity is assessed on a 5-point Likert scale from 1 (text is significantly more complex than the source) to 5 (text is significantly simpler than the source, or no further simplification was possible).  Note that sometimes you may see that the version which is supposed to be simplified is instead exactly identical to the source. In these cases, use your judgment, and if it is a very short sentence which cannot be simplified further, for example <em>"How are you?"</em>, then even if the text is not simpler than the source, it may be appropriate to assign a score of 5. A score of 3 is appropriate for cases where the text was not simplified nor made more complex, but potential for simplification exists. </li> <div style="height: 15px;" />  <div style="height: 10px;" />  </ul></ul> <strong>NOTE: please take a screeshot of this page for your future reference as you may not be able to access it again once you start the survey.</strong> <div style="height: 5px;" />  This one-hour study will start on the next page and comprises 25 pages.
> ````
>
> - HUMAN REVIEWERS INSTRUCTIONS FOR TEXT SUMMARIZATION
> ```html
> <big>Human evaluation of text summarization</big> <div style="height: 15px;" />IMPORTANT INFORMATION: We are conducting this research on the quality of text simplification, summarization, and grammatical error correction. This is the second study and looks at text summarization. It is not to be confused with the text simplification study you may have taken part in previously, as the two tasks are different. Here you will be asked to read some texts and their summarized versions and rate them according to the criteria specified below. Some of the content you will see is human generated, other is AI generated, but you will not be told which is which as this is a blind research. As the interface is not mobile / tablet friendly, we ask that you complete this study using a laptop or desktop. This summarization research will take about 8 hours, and you should only start working on this research if you can commit to do it in full. To make things simpler, we are splitting these 8 hours into 8 separate studies taking about one hour each, which you can take on separate days. All these 8 one-hour studies should be completed by the end of Thursday 11th May. This is study part 1 of 8. A link to the following 7 studies will be sent once you have accepted this commitment. Base pay is £12 per hour. Once you've completed the 8 studies, we will review the work, and if of a satisfactory quality, we will award a 10% discretionary bonus and invite you to take part to our last study which will look at grammatical error correction. This separate research will take another 4 hours and is optional, you can decide not to continue once you've completed the 8 one-hour studies on text summarization. Work will be reviewed as it comes through during UK working hours (excluding weekends and bank holidays). If at any point you have any questions or we have anything we'd like to communicate, feel free to message us though the Prolific messaging service, we will also use it if we need to get in touch, and potentially to discuss continuation of your work into the following studies. <div style="height: 15px;" /> <ul>  In the study you're about to start, which we estimate should take one hour, you will be shown 12 or 13 pieces of text taken mostly from older news articles. Alongside each of them, you will be shown 4 different possible summaries of the text. The purpose of text summarization is to make the text shorter and more concise without changing its overall meaning, omitting unimportant details while retaining the key content of the story. We ask that you carefully read the original text and rank each of the 4 summaries according to the following metrics, which are defined below. To ensure the best quality of outcome, we allocated roughly 5 minutes per page (note: this is significantly longer than we allowed for in the previous study on text simplification you may have taken, because the texts in this study are longer and more complex and we do not want you to feel rushed, as we value quality of output). <div style="height: 10px;" /> A summary box would look as follows (you will have 4 of these per source text). The definition of the criteria you will be asked to rate the summaries on is below. Please ensure you familiarise yourself with the criteria before proceeding. <head><style>* {margin: 0; padding: 0;} .imgbox {display: grid; height: 100%;} .center-fit { max-width: 100%; max-height: 100vh; margin: auto;}</style></head><body><div class="imgbox"><img class="center-fit" src='surveyflow/screenshot.png'></div></body>  <strong>NOTE: please take a screeshot of this page for your future reference as you may not be able to access it again once you start the survey.</strong><div style="height: 15px;" /> <ul>           <li>                <p><strong>Relevance</strong> </p>This assesses the quality of selection of important content from the source. The summary should include only important information from the source document. Please penalize summaries which contain redundancies and excess information. Relevance is assessed on a 5-point Likert scale from 1 (Not Relevant) to 5 (Super Relevant) </li> <div style="height: 15px;" /> <li> <p><strong>Fluency</strong></p>This assesses the quality of individual sentences. Sentences in the summary should have no formatting problems, capitalization errors or obviously ungrammatical sentences (e.g., fragments, missing components) that make the text difficult to read. Fluency is assessed on a 5-point Likert scale from 1 (Not Fluent) to 5 (Super Fluent)</li>  <div style="height: 15px;" /> <li> <p><strong>Coherence</strong></p> This assesses the collective quality of all sentences. The summary should be well-structured and well-organized. The summary should not just be a heap of related information, but should build from sentence to sentence to a coherent body of information about a topic. Coherence is assessed on a 5-point Likert scale from 1 (Not Coherent) to 5 (Super Coherent) </li> <div style="height: 15px;" /> <li> <p><strong>Consistency</strong></p> This assesses the factual alignment between the summary and the summarized source. A factually consistent summary contains only statements that are entailed by the source document. Summaries that contain hallucinated facts (information which is not present in the source document) should be penalized. Consistency is assessed on a 5-point Likert scale from 1 (Not Consistent) to 5 (Super Consistent) </li>        <div style="height: 10px;" />  </ul></ul> <strong>NOTE: please take a screeshot of this page for your future reference as you may not be able to access it again once you start the survey.</strong>
> ```
>
> - HUMAN REVIEWERS INSTRUCTIONS FOR GRAMMATICAL ERROR CORRECTION
> ```html
> <big>Human evaluation of Grammatical Error Correction</big> <div style="height: 15px;" />IMPORTANT INFORMATION: We are conducting this research on the quality of text simplification, summarization, and grammatical error correction (GEC). This is the third study and looks at GEC. Grammatical Error Correction (GEC) is the task of automatically detecting and correcting errors in text. The task not only includes the correction of grammatical errors, such as missing prepositions and mismatched subject-verb agreement, but also orthographic and semantic errors, such as misspellings and word choice errors respectively. You will be asked to read some sentences and their corrected versions and rate them according to the criteria specified below. Note that not all sentences include grammatical errors, if they do not, we would expect the corrected version to be identical to the source. Some of the content you will see is human generated, other is AI generated, but you will not be told which is which as this is a blind research. As the interface is not mobile / tablet friendly, we ask that you complete this study using a laptop or desktop. This GEC research will take about 4 hours, and you should only start working on this research if you can commit to do it in full. To make things simpler, we are splitting these 4 hours into 4 separate studies taking about one hour each, which you can take on separate days. All these 4 one-hour studies should be completed by the end of Wednesday 17th May. This is study part 1 of 4. A link to the following 3 studies will be sent once you have accepted this commitment. Base pay is £12 per hour. Once you've completed the 4 studies, we will review the work, and if of a satisfactory quality, we will award a 10% discretionary bonus.  <div style="height: 15px;" /> <ul>  In the study you're about to start, which we estimate should take one hour, you will be shown 25 pieces of text written by learners of English taken from an open source GEC dataset. Alongside each of them, you will be shown 4 different possible corrections of the text. The purpose of GEC is to rewrite the source sentence such that the corrected version is free of any grammatical errors which may be present in the source, but at the same time doing it with the minimum number of edits possible, without paraphrasing or making unnecessary syntax changes, unless absolutely necessary. We ask that you carefully read the original text and rank each of the 4 corrected versions according to the following metrics, which are defined below. <div style="height: 10px;" /> A GEC box would look as follows (you will have 4 of these per source text). The definition of the criteria you will be asked to rate the corrections on is below. Please ensure you familiarise yourself with the criteria before proceeding. <head><style>* {margin: 0; padding: 0;} .imgbox {display: grid; height: 100%;} .center-fit { max-width: 100%; max-height: 100vh; margin: auto;}</style></head><body><div class="imgbox"><img class="center-fit" src='surveyflow/screenshot.png'></div></body>  <strong>NOTE: please take a screeshot of this page for your future reference as you may not be able to access it again once you start the survey.</strong><div style="height: 15px;" /> <ul>           <li>                <p><strong>Semantics</strong> </p>This assesses whether the meaning of the text is preserved following the GEC. Semantic preservation is assessed on a 5-point Likert scale from 1 (Meaning Not Preserved) to 5 (Meaning fully preserved). NOTE: You should penalize corrections which change the meaning unnecessarily. For example, the sentence <em>"I wentt at Rome for my birthday"</em> should be corrected to <em>"I went to Rome for my birthday"</em>. A correction such as <em>"I went to Rome for my anniversary"</em> should be penalised in this category as they introduce unnecessary changes to the meaning. </li> <div style="height: 15px;" /> <li> <p><strong>Grammaticality</strong></p>This assesses the quality of the correction and answers the question "How many errors are left in the corrected sentence?". Please provide a count of the remaining errors, regardless of whether they were present in the source or they were newly introduced errors in the supposed corrected version.</li>  <div style="height: 15px;" /> <li> <p><strong>Over-correction</strong></p> Since there can be multiple ways to correct a sentence, this assesses whether the correction is unnecessarily verbose or makes unnecessary syntax changes. The best correction should be done with the minimum number of edits. For example, if the sentence <em>"I wentt at Rome for my birthday"</em> is corrected to <em>"I decided to go to Rome for my birthday"</em> this should be penalized under this category because it contains unnecessary syntax changes, even though the final sentence is grammatically correct. This metric answers the question: Is the system over-correcting or making unnecessary syntax changes? The answers should be "No", "Minor over-correction", "Moderate over-correction" or "Substantial over-correction".</li> <div style="height: 15px;" /> <div style="height: 10px;" />  </ul></ul> <strong>NOTE: please take a screeshot of this page for your future reference as you may not be able to access it again once you start the survey.</strong>
> ```
>
> - GPT4 PROMPT FOR SIMPLIFICATION
> ```
>  You're GPT4 and are about to start a task where you will be shown some pieces of text taken mostly from older news articles, alongside 4 different possible simplified versions of each text, and you will be asked to evaluate the quality of the simplifications based on some metrics defined below. The purpose of text simplification can be varied, but include things such as making the text easier to read, more accessible to language learners, or explaining a medical or legal text in plain speak. We ask that you carefully read the original text and rank each of the 4 simplifications according to the following metrics, which are defined below.
>
>
> Semantics, or adequacy: This assesses whether the meaning of the text is preserved further to the simplification. Semantic preservation is assessed on a 5-point Likert scale from 1 (Meaning Not Preserved) to 5 (Meaning fully preserved). NOTE: You should penalize simplified text which is not self contained. For example, if the sentence "Scientists have no idea how eels protect their own brain and muscles from being damaged by electricity" is simplified as "Scientists have no idea how they stay safe", this should be rated around 2 out of 5 as it is not possible by just reading the simplified version to understand what "they" is referring to. Like wise if the simplified text contains hallucinations (facts not contained in the source) it should be penalised. For example if "He yelled" is simplified as "His father Russell shouted" this should be rated around 2 under this category. The omission of some unimportant details from the source into the simplified version is acceptable at your discretion, if the text meaning is preserved.
>
>
> Grammaticality (or fluency). This assesses whether the simplified text remains grammatical and understandable. Sentences should have no formatting problems, capitalization errors or obviously ungrammatical sentences (e.g., fragments, missing components) that make the text difficult to read. Grammaticality / Fluency is assessed on a 5-point Likert scale from 1 (Not Fluent) to 5 (Super Fluent)
>
>
> Simplicity. This assesses whether the simplified text, considered in its entirety, is simpler than the source text. Simplicity is assessed on a 5-point Likert scale from 1 (text is significantly more complex than the source) to 5 (text is significantly simpler than the source, or no further simplification was possible). Note that sometimes you may encounter very short sentences which cannot be simplified further and are repeated identically, for example "How are you?". In such cases, even if the text is not simpler than the source, it may be appropriate to assign a score of 5. A score of 3 is appropriate for cases where the text was not simplified nor made more complex, but potential for simplifcation exists.
>
>
> We will pass you the input you need to rank in json format
> Please reply with the scores in json format.
> This is an example json query where "original_input" is the source text, 'id' is the unique identifier, and all other keys represent output texts which you need to evaluate.
> {"original_input": "He is the first to market a simplified version of the tool to the public , a project that , for now , is not dangerous .", "gold_reference": "Zayner is the first to sell a gene-changing tool to regular people . His set is not dangerous .", "google/flan-t5-xxl": "He is the first to market a simplified version of the tool to the public, a project that, for now, is not dangerous.", "davinci-instruct-beta": "He is the first to market a simplified version of the tool to the public.\n\nHe is the first to market a simplified version of the tool to the public.", "gpt-3.5-turbo": "He created a safer version of the tool and introduced it to the public before anyone else.", "id": "9ec83328fbea94d7cdc4b92aba92f64abaa8727f6c0c99cc3b588b58bb8bb3f5"}
>
>
> Your answer should contain the id and the scores, for example, if you wish to give gold_reference a semantics score of 2, a grammaticality score of 5 and a simplicity score of 4, and you wish to give google/flan-t5-xxl a semantics score of 5, a grammaticality score of 5, a simplicity score of 4, and you wish to give davinci-instruct-beta a semantics score of 5, a grammaticality score of 5, a simplicity score of 4, and you wish to give gpt-3.5-turbo a semantics score of 5, a grammaticality score of 5 and a simplicity score of 5, then you should return the following output (note how the id item needs to be preserved to allowed for identification):
> {"gold_reference": {"semantics": 2, "grammaticality": 5, "simplicity": 4}, "google/flan-t5-xxl": {"semantics": 5, "grammaticality": 5, "simplicity": 4}, "davinci-instruct-beta": {"semantics": 5, "grammaticality": 5, "simplicity": 4}, "gpt-3.5-turbo": {"semantics": 5, "grammaticality": 5, "simplicity": 5}, "id": "9ec83328fbea94d7cdc4b92aba92f64abaa8727f6c0c99cc3b588b58bb8bb3f5"}
>
>
> Is this clear? Do you have any questions or are you ready to start?
> ```
>
> - GPT4 PROMPT FOR SUMMARIZATION
> ```html
> You're GPT4 and are about to start a task where you will be shown some pieces of text taken mostly from older news articles, alongside 4 different possible summaries of each text, and you will be asked to evaluate the quality of the summaries based on some metrics defined below. The purpose of text summarization is to make the text shorter and more concise without changing its overall meaning, omitting unimportant details while retaining the key content of the story. We ask that you carefully read the original text and rank each of the 4 summaries according to the following metrics, which are defined below.
>
>
> Relevance. This assesses the quality of selection of important content from the source. The summary should include only important information from the source document. Please penalize summaries which contain redundancies and excess information. Relevance is assessed on a 5-point Likert scale from 1 (Not Relevant) to 5 (Super Relevant)
>
>
> Fluency. This assesses the quality of individual sentences. Sentences in the summary should have no formatting problems, capitalization errors or obviously ungrammatical sentences (e.g., fragments, missing components) that make the text difficult to read. Fluency is assessed on a 5-point Likert scale from 1 (Not Fluent) to 5 (Super Fluent)
>
>
> Coherence. This assesses the collective quality of all sentences. The summary should be well-structured and well-organized. The summary should not just be a heap of related information, but should build from sentence to sentence to a coherent body of information about a topic. Coherence is assessed on a 5-point Likert scale from 1 (Not Coherent) to 5 (Super Coherent)
>
>
> Consistency. This assesses the factual alignment between the summary and the summarized source. A factually consistent summary contains only statements that are entailed by the source document. Summaries that contain hallucinated facts (information which is not present in the source document) should be penalized. Consistency is assessed on a 5-point Likert scale from 1 (Not Consistent) to 5 (Super Consistent)
>
>
> We will pass you the input you need to rank in json format.
> Please reply with the scores in json format.
> This is an example json query where "original_input" is the source text, "id" is the unique identifier, and all other keys represent output texts which you need to evaluate.
> {"original_input": "(CNN)Love it or hate it, Jared Leto's interpretation of the Joker is an internet sensation. The Oscar winner put on white makeup (and a lot of tattoos this time) to portray the Clown Prince of Crime in the upcoming movie \"Suicide Squad.\" Set for release August 5, 2016, \"Suicide Squad\" is based on the DC Comics series and also stars Will Smith, Margot Robbie and Viola Davis. Twitter users got their first look at Leto in character Friday night, and the memes started almost immediately. From comparisons to \"Home Alone\" to an imagining of Ben Affleck tatted up, people on social media put their Photoshopping skills to work all weekend. Which is your favorite?", "gold_reference": "Leto will play the Clown Prince of Crime in 2016's \"Suicide Squad\"\nThe first picture of Leto in character led to a series of spoof photos .", "bigscience/T0pp": "The internet is buzzing about Jared Leto's new look as the Joker. The Oscar winner is playing the Clown Prince of Crime in \"Suicide Squad\"", "text-davinci-003": " Jared Leto's interpretation of the Joker in the upcoming movie \"Suicide Squad\" has become an internet sensation. The movie, set for release August 5, 2016, also stars Will Smith, Margot Robbie and Viola Davis. Twitter users got their first look at Leto in character Friday night, and the memes started almost immediately, with people Photoshopping him in various ways.", "gpt-3.5-turbo": "Jared Leto's portrayal of the Joker in the upcoming movie \"Suicide Squad\" has become an internet sensation. Twitter users have been sharing memes and Photoshopped images of Leto's character, who is heavily tattooed, since his first appearance in character on Friday. The movie is set for release on August 5, 2016 and also stars Will Smith, Margot Robbie, and Viola Davis.", "id": "49f45c17ee34e2aa420ef557d944a6f2df96f5226c2793e4a9d379c517fde9a5"}
>
>
> Your answer should contain the id and the scores, for example, using the example given above, if you wish to give gold_reference a relevance score of 2, a fluency score of 5, a coherence score of 4 and a consistency score of 4, and you wish to give bigscience/T0pp a relevance score of 3, a fluency score of 5, a coherence score of 1 and a consistency score of 3,  and you wish to give text-davinci-003 a relevance score of 5, a fluency score of 4, a coherence score of 3 and a consistency score of 4, and you wish to give gpt-3.5-turbo a relevance score of 4, a fluency score of 3, a coherence score of 5 and a consistency score of 3, then you should return the following output (note how the id item needs to be preserved to allow for identification):
> {"gold_reference": {"relevance": 2, "fluency": 5, "coherence": 4, "consistency": 4}, "bigscience/T0pp": {"relevance": 3, "fluency": 5, "coherence": 1, "consistency": 3}, "text-davinci-003": {"relevance": 5, "fluency": 4, "coherence": 3, "consistency": 4}, "gpt-3.5-turbo": {"relevance": 4, "fluency": 3, "coherence": 5, "consistency": 3}, "id": "49f45c17ee34e2aa420ef557d944a6f2df96f5226c2793e4a9d379c517fde9a5"}
>
>
> Is this clear? Do you have any questions or are you ready to start?
> ```
>
> - GPT4 PROMPT FOR GRAMMATICAL ERROR CORRECTION
> ```html
> You're GPT4 and are about to start a task where you will be shown some sentences written by learners of English. Some of these sentences will contain errors, and alongside each sentence you will be shown 4 different possible corrections, and you will be asked to evaluate the quality of the correction based on some metrics defined below. This task is called Grammatical Error Correction (GEC), and is the task of automatically detecting and correcting errors in text. The task not only includes the correction of grammatical errors, such as missing prepositions and mismatched subject-verb agreement, but also orthographic and semantic errors, such as misspellings and word choice errors respectively. Note that not all sentences you will see include grammatical errors; if they do not, we would expect the corrected version to be identical to the source. We ask that you carefully read the original sentence and rank each of the 4 corrections according to the following metrics, which are defined below.
>
>
> Semantics. This assesses whether the meaning of the text is preserved following the GEC. Semantic preservation is assessed on a 5-point Likert scale from 1 (Meaning Not Preserved) to 5 (Meaning fully preserved). NOTE: You should penalize corrections which change the meaning unnecessarily. For example, the sentence "I wentt at Rome for my birthday" should be corrected to "I went to Rome for my birthday". A correction such as "I went to Rome for my anniversary" should be penalised in this category as it introduces unnecessary changes to the meaning.
>
>
> Grammaticality. This assesses the quality of the correction and answers the question "How many errors are left in the corrected sentence?". Please provide a count of the remaining errors, regardless of whether they were present in the source or they were newly introduced errors in the supposed corrected version. The options are "0", "1", "2 or more". Note that, unlike for semantics where a score of 5 is better than a score of 1, here a score of "0" is better than a score of "1" which is better than a score of "2 or more" (this is because if there are 0 errors remaining, the GEC task has been fulfilled).
>
>
> Over-correction. Since there can be multiple ways to correct a sentence, this assesses whether the correction is unnecessarily verbose or makes unnecessary syntax changes. The best correction should be done with the minimum number of edits. For example, if the sentence "I wentt at Rome for my birthday" is corrected to "I decided to go to Rome for my birthday" this should be penalized under this category because it contains unnecessary syntax changes, even though the final sentence is grammatically correct. This metric answers the question: Is the system over-correcting or making unnecessary syntax changes? The answers should be "No", "Minor over-correction", "Moderate over-correction" or "Substantial over-correction".
>
>
> We will pass you the input you need to rank in json format.
> Please reply with the scores in json format.
> This is an example json query where "original_input" is the source sentence, "id" is the unique identifier, and all other keys represent the output corrected sentences which you need to evaluate.
> {"original_input": "Travalling by car is also much more convenient.", "gold_reference": "Travelling by car is also much more convenient.", "opt-iml-max-30b": "Travalling by car is also much more convenient.", "text-davinci-003": "Traveling by car is also much more convenient.", "gpt-3.5-turbo": "Traveling by car is also much more convenient.", "id": "765d0adf8a8c58e2617a435be0d2451edcfcb782492ac662432da53843607d9a"}
>
>
> Your answer should contain the id and the scores, for example, using the example given above, if you wish to give gold_reference a semantics score of 5, a grammaticality score of "0", an overcorrection score of "No", and you wish to give opt-iml-max-30b a semantics score of 4, a grammaticality score of "1", an overcorrection score of "Minor over-correction",  and you wish to give text-davinci-003 a semantics score of 3, a grammaticality score of "2 or more", an overcorrection score of "Substantial over-correction", and you wish to give gpt-3.5-turbo a semantics score of 5, a grammaticality score of "0", an overcorrection score of "Moderate over-correction", then you should return the following output (note how the id item needs to be preserved to allow for identification):
> {"gold_reference": {"semantics": 5, "grammaticality": "0", "overcorrection": "No"}, "opt-iml-max-30b": {"semantics": 4, "grammaticality": "1", "overcorrection": "Minor over-correction"}, "text-davinci-003": {"semantics": 3, "grammaticality": "2 or more", "overcorrection": "Substantial over-correction"}, "gpt-3.5-turbo": {"semantics": 5, "grammaticality": "0", "overcorrection":  "Moderate over-correction"}, "id": "49f45c17ee34e2aa420ef557d944a6f2df96f5226c2793e4a9d379c517fde9a5"}
>
>
> Is this clear? Do you have any questions or are you ready to start?
> ```

---

### Official Review · Reviewer_Etgw · 2023-08-03

**Soundness:** 3

**Excitement:**

3: Ambivalent: It has merits (e.g., it reports state-of-the-art results, the idea is nice), but there are key weaknesses (e.g., it describes incremental work), and it can significantly benefit from another round of revision. However, I won't object to accepting it if my co-reviewers champion it.

**Missing References:**

[2] Cheng-Han Chiang and Hung-yi Lee. 2023. Can Large Language Models Be an Alternative to Human Evaluations?. In Proceedings of the 61st Annual Meeting of the Association for Computational Linguistics (Volume 1: Long Papers), pages 15607–15631, Toronto, Canada. Association for Computational Linguistics.

[3] Liu, Yang, et al. "Gpteval: Nlg evaluation using gpt-4 with better human alignment." arXiv preprint arXiv:2303.16634 (2023).

[4] Fu, Jinlan, et al. "Gptscore: Evaluate as you desire." arXiv preprint arXiv:2302.04166 (2023).

[2] is concurrent to the current paper, but it is highly relevant and should be cited according to the author's guidelines.
[3] is highly relevant, it also evaluates summarization tasks.

**Paper Topic And Main Contributions:**

Paper Topic
===
This paper evaluates the performance of several LLMs (FLAN-T5, FLAN-UL2, T0pp, OPT-IML, GPT3, InstructGPT, and ChatGPT) on three sequence-to-sequence tasks: text summarization, text simplification, and grammar error correction. They show that automatic evaluation metrics relying on the gold reference cannot give a model ranking consistent with human evaluation rankings. They attribute this to low-quality references. They also show that the output of LLMs mostly outperforms the gold reference. Last, they show that using GPT-4 to rate the LLM outputs can yield model rankings that are highly consistent with the ranking provided by human evaluators.

Contribution
===
NLP engineering experiment

**Questions For The Authors:**

A. If FLAN-T5 copies the input in text classification most of the time, shouldn't it be rated as the most semantic preserving by GPT4?

**Reasons To Accept:**

- This paper has clear and important takeaways: automatic evaluation metrics are insufficient for evaluating LLM outputs; GPT-4 is a good evaluator; ground truth references are bad.
- This paper shows that GPT4 is not very good as an evaluator for grammar error correction. Such a negative result can help future researchers avoid using it and draw false conclusions.

**Reasons To Reject:**

- **Lack of details of experiment settings**
    - The paper does not show or discuss how GPT-4 was prompted to rate the samples. Whether the instructions given to GPT-4 are identical to those given to human evaluators is not revealed, so it is hard to judge if the difference between human's ranking and GPT-4's ranking is due to instruction difference. Based on my experience, those instructions can have a significant influence on the results.
    - The number of samples rated by GPT-4 and whether they overlapped with the samples used in the human evaluation are not clearly stated. I assume that the samples should overlap since the paper calculates Krippendorff $\alpha$.
    - The version of OpenAI models (or the time of prompting those models) is not specified.
    - The instructions given to the human evaluators are not specified. The interface is not shown. This raises some questions: are the order of outputs from the models and ground truth fixed in human evaluation? Are humans informed that one of the texts is written by humans and others are model generated?
- **Odd experiment settings**
    - The models used for seq2seq generation are very strange. Precisely, the **GPT3** (`text-davinci-003`) and **InstructGPT** (`davinci-instruct-beta`). In fact, `text-davinci-003` is an InstructGPT model; it is fine-tuned using RLHF. Also, there is currently no `davinci-instruct-beta` on OpenAI now. Please refer to the [OpenAI documentation](https://platform.openai.com/docs/models]). Here, it is unclear which models are used in the paper, undermining reproducibility.
- **Limited novelty**
    - Automatic evaluation metrics is not well-correlated with human is a well-known issue; see references [55, 45, 6, 33] in [1]. (This is listed as the first contribution in this paper)
    - Low-quality reference in summarization is also well-known; see the SummEval paper and [1]. (This is listed as the second contribution in this paper.)
    - The fact that GPT4 (or GPT3.5) can be used for automatic evaluation in NLG is shown in many prior works, and I am just listing two of them: [2], [3], and [4]. [2] and [3] are concurrent, but [4] is not.


[1] Stiennon, Nisan, et al. "Learning to summarize with human feedback." Advances in Neural Information Processing Systems 33 (2020): 3008-3021.

[2] Cheng-Han Chiang and Hung-yi Lee. 2023. Can Large Language Models Be an Alternative to Human Evaluations?. In Proceedings of the 61st Annual Meeting of the Association for Computational Linguistics (Volume 1: Long Papers), pages 15607–15631, Toronto, Canada. Association for Computational Linguistics.

[3] Liu, Yang, et al. "Gpteval: Nlg evaluation using gpt-4 with better human alignment." arXiv preprint arXiv:2303.16634 (2023).

[4] Fu, Jinlan, et al. "Gptscore: Evaluate as you desire." arXiv preprint arXiv:2302.04166 (2023).

**Reproducibility:**

2: Would be hard pressed to reproduce the results. The contribution depends on data that are simply not available outside the author's institution or consortium; not enough details are provided.

**Reviewer Confidence:**

5: Positive that my evaluation is correct. I read the paper very carefully and I am very familiar with related work.

**Typos Grammar Style And Presentation Improvements:**

Title: Sequence-to-Sequence
Line 349 is an incomplete sentence
Line 217: **was** scored
Line 153 do not need the comma

---

> ### Author Rebuttal · Authors · 2023-08-29
>
> We thank Reviewer 2 for the comments, to which we have given a lot of thought. Please find below our responses to their concerns and questions in the same order as they were raised, and we hope the reviewer finds them clear.
>
> **Lack of details of experiment settings**
> - On the first point regarding lack of details on GPT4 prompting, as specified in lines 155-158, we included the full evaluation criteria and their assessment scales in the prompt. It is true that we did not report the prompts verbatim in the paper. This is because the prompts were very long. The instructions given the human reviewers and GPT-4 were almost identical, with minor differences (for example, in the human reviewer version we added details on time to spend and pay rate, whereas in the GPT4 prompt we included details on how to return the output in a json format so we could easily parse it). For the avoidance of doubt, we have included all instructions given to human reviewers and GPT4 at the end of this rebuttal, and are happy to add them to an Appendix in the camera-ready version.
> - On the point about lack of clarity on whether the samples rated by GPT-4 and human reviewers overlap, we can confirm that all three human annotators and GPT-4 annotated the exact same set of 100 samples for each of the three tasks. We appreciate this was not explicitly stated in the text, and will make that clearer in the camera-ready version.
> - At the time of writing the paper, we used the latest available version of ChatGPT. Most experiments were run in the first half of 2023. Using `gpt-3.5-turbo` in the API will always yield the latest version, I believe the version we used at the time is the one currently labelled as `gpt-3.5-turbo-0301` which was last updated on 1st March 2023. See here for more details https://platform.openai.com/docs/models/gpt-4. As this is a very rapidly changing landscape, newer versions have since become available. Regarding other openAI models, such as `text-davinci-003` and `davinci-instruct-beta`, they do not seem to have changed since the experiments were run, and as far as we are aware they are not currently versioned.
> - We confirm that the order of appearance of the outputs from the models and the gold reference are fixed in the human evaluation questionnaire, and are also the same in the GPT4 evaluation prompts. Human annotators were informed that one of the four outputs was human-generated, but were not told which one. The human evaluation survey for each of the three tasks had a landing page which contained detailed information about the task and evaluation criteria. Once the reviewer has familiarized themselves they click on a “Proceed” button which takes them to the first sample. Each sample was on a different webpage to avoid confusion, and the users were able to move between samples using browser buttons. On each page, the main input text was shown at the top. The user was then shown four different outputs below each input, and below each output they were asked to rate the output above using the task-specific criteria outlined in the paper. They were able to see all four outputs for the same input on the same webpage, making it easier to give scores which were correct relative to each other. The detailed instructions given are very long and are attached at the end of this rebuttal. As mentioned above, we are happy to add them to an Appendix in the camera-ready version.
>
> **Odd experiment settings**
> - According to OpenAI, `text-davinci-003`’s strengths are “Complex intent, cause and effect, creative generation, search, summarization for audience” and `davinci-instruct-beta`’s strengths are “complex intent, cause and effect”. While we agree that they are similar models, they are not the same model; they indeed show different results. It is true that `text-davinci-003`, which is always mapped to the best model of the GPT3 series, is currently an InstructGPT model, but it wasn’t always the case. It is not correct to say there is no `davinci-instruct-beta` model on OpenAI. However, we have just noticed that `davinci-instruct-beta` has recently been marked as a “legacy” model, which wasn’t the case a few months ago, and therefore may be hidden from the main webpage. In order to find this model on the the OpenAI playground, you can navigate here https://platform.openai.com/playground?mode=complete&model=davinci-instruct-beta and select Mode → “Complete” (not the default of “Chat”) and on the “Models” dropdown, click “show more models”, where you will be able to see `davinci-instruct-beta` amongst the available models. We will make this clearer in the camera-ready version.
>
> **Limited novelty**
>
> We are aware that poor correlation between automatic metrics and human judgment, as well as the issue of low quality gold references, have been raised before in similar and different contexts. However, we believe we add a significant contribution on evaluating automatic evaluation metrics capability to capture the performance of LLMs, some of which have been recently released, as well as using GPT4 as an evaluator for such models. We have reviewed the referenced work, and we note that [2] explores LLMs evaluation on open-ended story generation and adversarial attacks, which are different tasks to the ones we analysed. One of the questions posed is _"How grammatically correct is the text of the story fragment? (on a scale of 1-5, with 1 being the lowest?)"_, which is similar to one of the evaluation criteria we used, but we also looked at many others. Reference [3] assesses the quality of LLMs outputs on text summarization and dialogue generation, whereas [4] runs experiments on four text generation tasks, namely dialogue response generation, text summarization, data-to-text, and machine translation, with very similar evaluation criteria to the ones we used on text summarization (Table 1 in [4]). However, none of these referenced works focus on text simplification or grammatical error correction.  As such we believe the particulars of our work differ from previous and concurrent works and our paper provides new insights, despite some overlap in the general research topic. However, we thank the reviewer for kindly pointing out relevant concurrent and recent work, in particular references [2], [3] and [4] and we would be happy to include those citations in our camera-ready version.
>
> **Questions for the authors**
> - **Question 1**: We observed that GPT4 rated Flan-T5 as second best model for semantic preservation on text simplification. In terms of absolute scores, as it can be seen in Table 6 (Appendix E), GPT4 gave ChatGPT an average score of 4.66 for semantic preservation, and it gave Flan-T5 a score of 4.55, so the absolute difference is not large. We agree with the reviewer that since Flan-T5 copies the input in text simplification most of the time, it should be rated as the most semantic preserving by GPT4, as it was by human annotators. We do not know why this was not the case, with GPT4 being a paid-for black-box model, and we only limited ourselves to reporting the observed results. An investigation into possible explanations behind this behaviour of GPT-4 would certainly be interesting, but is currently beyond the scope of this paper.
>
>
> On a final note on reproducibility, we hope that the provision of the full instructions below will help alleviate the concerns. We made the decision not to include them in the paper due to their length, but are happy to do so in an Appendix in the camera-ready version.
>
> Below are the instructions given to human reviewers and GPT4 by task, as requested. The human reviewers instructions contain html tags as they were delivered as a UI.
>
> - HUMAN REVIEWERS INSTRUCTIONS FOR TEXT SIMPLIFICATION
>
> ```html
> <big>Human evaluation of text simplification</big> <div style="height: 15px;" />IMPORTANT INFORMATION: We are conducting this research on the quality of text simplification, summarization, and grammatical error correction. We will start from text simplification. You will be asked to read some short texts and their simplified versions and rate them according to some criteria which are specified below. Some of the content you will see is human generated, other is AI generated, but you will not be told which is which as this is a blind research. As the interface is not mobile / tablet friendly, we ask that you complete this study using a laptop or desktop. This research is divided into 3 parts taking about 4, 8 and 4 hours respectively (around 16 hours in total). You should only start working on this research if you can at least commit for the first part (4 hours). To make things simpler, we are splitting these 4 hours into 4 separate studies taking about one hour each, which you can take on separate days. All these 4 one-hour studies should be completed by the end of Thursday 4th May. This is study part 1 of 4. A link to the following 3 studies will be sent once you have accepted this commitment. Base pay is £12 per hour. Once you've completed the first 4 one-hour studies, we will review the work, and if of a satisfactory quality, we will award a 10% discretionary bonus and invite you to take part to the two remaining parts of the study which will require a longer time commitment. You can decide not to continue with the research after you have completed the first 4 one-hour studies. Work will be reviewed as it comes through during UK working hours (excluding weekends and bank holidays). If at any point you have any questions or we have anything we'd like to communicate, feel free to message us through the Prolific messaging service, we will also use it if we need to get in touch, and potentially to discuss continuation of your work into the following studies. <div style="height: 15px;" /> <ul>  In the study you're about to start, which we estimate should take one hour, you will be shown 25 pieces of text taken mostly from older news articles. Alongside each of them, you will be shown 4 different possible simplified versions of the text. The purpose of text simplification can be varied, but include things such as making the text easier to read, more accessible to language learners, or explaining a medical or legal text in plain speak. We ask that you carefully read the original text and rank each of the 4 simplifications according to the following metrics, which are defined below. <div style="height: 10px;" /> A simplification box would look as follows (you will have 4 of these per source text) <head><style>* {margin: 0; padding: 0;} .imgbox {display: grid; height: 100%;} .center-fit { max-width: 100%; max-height: 100vh; margin: auto;}</style></head><body><div class="imgbox"><img class="center-fit" src='surveyflow/screenshot.png'></div></body>  <strong>NOTE: please take a screeshot of this page for your future reference as you may not be able to access it again once you start the survey.</strong><div style="height: 15px;" /> <ul>           <li>                <p><strong>Semantics (or adequacy)</strong> </p> This assesses whether the meaning of the text is preserved further to the simplification. Semantic preservation is assessed on a 5-point Likert scale from 1 (Meaning Not Preserved) to 5 (Meaning fully preserved). NOTE: You should penalize simplified text which is not self contained. For example, if the sentence <em>"Scientists have no idea how eels protect their own brain and muscles from being damaged by electricity"</em> is simplified as <em>"Scientists have no idea how they stay safe"</em>, this should be rated around 2 out of 5 as it is not possible by just reading the simplified version to understand what "they" is referring to. Like wise if the simplified text contains hallucinations (facts not contained in the source) it should be penalised. For example if <em>"He yelled"</em> is simplified as <em>"His father Russell shouted"</em> this should be rated around 2 under this category. The omission of some unimportant details from the source into the simplified version is acceptable at your discretion, if the text meaning is preserved. </li> <div style="height: 15px;" /> <li> <p><strong>Grammaticality (or fluency)</strong></p>This assesses whether the simplified text remains grammatical and understandable. Sentences should have no formatting problems, capitalization errors or obviously ungrammatical sentences (e.g., fragments, missing components) that make the text difficult to read. Grammaticality / Fluency is assessed on a 5-point Likert scale from 1 (Not Fluent) to 5 (Super Fluent)</li>  <div style="height: 15px;" /> <li> <p><strong>Simplicity</strong></p> This assesses whether the simplified text, considered in its entirety, is simpler than the source text. Simplicity is assessed on a 5-point Likert scale from 1 (text is significantly more complex than the source) to 5 (text is significantly simpler than the source, or no further simplification was possible).  Note that sometimes you may see that the version which is supposed to be simplified is instead exactly identical to the source. In these cases, use your judgment, and if it is a very short sentence which cannot be simplified further, for example <em>"How are you?"</em>, then even if the text is not simpler than the source, it may be appropriate to assign a score of 5. A score of 3 is appropriate for cases where the text was not simplified nor made more complex, but potential for simplification exists. </li> <div style="height: 15px;" />  <div style="height: 10px;" />  </ul></ul> <strong>NOTE: please take a screeshot of this page for your future reference as you may not be able to access it again once you start the survey.</strong> <div style="height: 5px;" />  This one-hour study will start on the next page and comprises 25 pages.
> ````
>
>
> - HUMAN REVIEWERS INSTRUCTIONS FOR TEXT SUMMARIZATION
> ```html
> <big>Human evaluation of text summarization</big> <div style="height: 15px;" />IMPORTANT INFORMATION: We are conducting this research on the quality of text simplification, summarization, and grammatical error correction. This is the second study and looks at text summarization. It is not to be confused with the text simplification study you may have taken part in previously, as the two tasks are different. Here you will be asked to read some texts and their summarized versions and rate them according to the criteria specified below. Some of the content you will see is human generated, other is AI generated, but you will not be told which is which as this is a blind research. As the interface is not mobile / tablet friendly, we ask that you complete this study using a laptop or desktop. This summarization research will take about 8 hours, and you should only start working on this research if you can commit to do it in full. To make things simpler, we are splitting these 8 hours into 8 separate studies taking about one hour each, which you can take on separate days. All these 8 one-hour studies should be completed by the end of Thursday 11th May. This is study part 1 of 8. A link to the following 7 studies will be sent once you have accepted this commitment. Base pay is £12 per hour. Once you've completed the 8 studies, we will review the work, and if of a satisfactory quality, we will award a 10% discretionary bonus and invite you to take part to our last study which will look at grammatical error correction. This separate research will take another 4 hours and is optional, you can decide not to continue once you've completed the 8 one-hour studies on text summarization. Work will be reviewed as it comes through during UK working hours (excluding weekends and bank holidays). If at any point you have any questions or we have anything we'd like to communicate, feel free to message us though the Prolific messaging service, we will also use it if we need to get in touch, and potentially to discuss continuation of your work into the following studies. <div style="height: 15px;" /> <ul>  In the study you're about to start, which we estimate should take one hour, you will be shown 12 or 13 pieces of text taken mostly from older news articles. Alongside each of them, you will be shown 4 different possible summaries of the text. The purpose of text summarization is to make the text shorter and more concise without changing its overall meaning, omitting unimportant details while retaining the key content of the story. We ask that you carefully read the original text and rank each of the 4 summaries according to the following metrics, which are defined below. To ensure the best quality of outcome, we allocated roughly 5 minutes per page (note: this is significantly longer than we allowed for in the previous study on text simplification you may have taken, because the texts in this study are longer and more complex and we do not want you to feel rushed, as we value quality of output). <div style="height: 10px;" /> A summary box would look as follows (you will have 4 of these per source text). The definition of the criteria you will be asked to rate the summaries on is below. Please ensure you familiarise yourself with the criteria before proceeding. <head><style>* {margin: 0; padding: 0;} .imgbox {display: grid; height: 100%;} .center-fit { max-width: 100%; max-height: 100vh; margin: auto;}</style></head><body><div class="imgbox"><img class="center-fit" src='surveyflow/screenshot.png'></div></body>  <strong>NOTE: please take a screeshot of this page for your future reference as you may not be able to access it again once you start the survey.</strong><div style="height: 15px;" /> <ul>           <li>                <p><strong>Relevance</strong> </p>This assesses the quality of selection of important content from the source. The summary should include only important information from the source document. Please penalize summaries which contain redundancies and excess information. Relevance is assessed on a 5-point Likert scale from 1 (Not Relevant) to 5 (Super Relevant) </li> <div style="height: 15px;" /> <li> <p><strong>Fluency</strong></p>This assesses the quality of individual sentences. Sentences in the summary should have no formatting problems, capitalization errors or obviously ungrammatical sentences (e.g., fragments, missing components) that make the text difficult to read. Fluency is assessed on a 5-point Likert scale from 1 (Not Fluent) to 5 (Super Fluent)</li>  <div style="height: 15px;" /> <li> <p><strong>Coherence</strong></p> This assesses the collective quality of all sentences. The summary should be well-structured and well-organized. The summary should not just be a heap of related information, but should build from sentence to sentence to a coherent body of information about a topic. Coherence is assessed on a 5-point Likert scale from 1 (Not Coherent) to 5 (Super Coherent) </li> <div style="height: 15px;" /> <li> <p><strong>Consistency</strong></p> This assesses the factual alignment between the summary and the summarized source. A factually consistent summary contains only statements that are entailed by the source document. Summaries that contain hallucinated facts (information which is not present in the source document) should be penalized. Consistency is assessed on a 5-point Likert scale from 1 (Not Consistent) to 5 (Super Consistent) </li>        <div style="height: 10px;" />  </ul></ul> <strong>NOTE: please take a screeshot of this page for your future reference as you may not be able to access it again once you start the survey.</strong>
> ```
>
> - HUMAN REVIEWERS INSTRUCTIONS FOR GRAMMATICAL ERROR CORRECTION
> ```html
> <big>Human evaluation of Grammatical Error Correction</big> <div style="height: 15px;" />IMPORTANT INFORMATION: We are conducting this research on the quality of text simplification, summarization, and grammatical error correction (GEC). This is the third study and looks at GEC. Grammatical Error Correction (GEC) is the task of automatically detecting and correcting errors in text. The task not only includes the correction of grammatical errors, such as missing prepositions and mismatched subject-verb agreement, but also orthographic and semantic errors, such as misspellings and word choice errors respectively. You will be asked to read some sentences and their corrected versions and rate them according to the criteria specified below. Note that not all sentences include grammatical errors, if they do not, we would expect the corrected version to be identical to the source. Some of the content you will see is human generated, other is AI generated, but you will not be told which is which as this is a blind research. As the interface is not mobile / tablet friendly, we ask that you complete this study using a laptop or desktop. This GEC research will take about 4 hours, and you should only start working on this research if you can commit to do it in full. To make things simpler, we are splitting these 4 hours into 4 separate studies taking about one hour each, which you can take on separate days. All these 4 one-hour studies should be completed by the end of Wednesday 17th May. This is study part 1 of 4. A link to the following 3 studies will be sent once you have accepted this commitment. Base pay is £12 per hour. Once you've completed the 4 studies, we will review the work, and if of a satisfactory quality, we will award a 10% discretionary bonus.  <div style="height: 15px;" /> <ul>  In the study you're about to start, which we estimate should take one hour, you will be shown 25 pieces of text written by learners of English taken from an open source GEC dataset. Alongside each of them, you will be shown 4 different possible corrections of the text. The purpose of GEC is to rewrite the source sentence such that the corrected version is free of any grammatical errors which may be present in the source, but at the same time doing it with the minimum number of edits possible, without paraphrasing or making unnecessary syntax changes, unless absolutely necessary. We ask that you carefully read the original text and rank each of the 4 corrected versions according to the following metrics, which are defined below. <div style="height: 10px;" /> A GEC box would look as follows (you will have 4 of these per source text). The definition of the criteria you will be asked to rate the corrections on is below. Please ensure you familiarise yourself with the criteria before proceeding. <head><style>* {margin: 0; padding: 0;} .imgbox {display: grid; height: 100%;} .center-fit { max-width: 100%; max-height: 100vh; margin: auto;}</style></head><body><div class="imgbox"><img class="center-fit" src='surveyflow/screenshot.png'></div></body>  <strong>NOTE: please take a screeshot of this page for your future reference as you may not be able to access it again once you start the survey.</strong><div style="height: 15px;" /> <ul>           <li>                <p><strong>Semantics</strong> </p>This assesses whether the meaning of the text is preserved following the GEC. Semantic preservation is assessed on a 5-point Likert scale from 1 (Meaning Not Preserved) to 5 (Meaning fully preserved). NOTE: You should penalize corrections which change the meaning unnecessarily. For example, the sentence <em>"I wentt at Rome for my birthday"</em> should be corrected to <em>"I went to Rome for my birthday"</em>. A correction such as <em>"I went to Rome for my anniversary"</em> should be penalised in this category as they introduce unnecessary changes to the meaning. </li> <div style="height: 15px;" /> <li> <p><strong>Grammaticality</strong></p>This assesses the quality of the correction and answers the question "How many errors are left in the corrected sentence?". Please provide a count of the remaining errors, regardless of whether they were present in the source or they were newly introduced errors in the supposed corrected version.</li>  <div style="height: 15px;" /> <li> <p><strong>Over-correction</strong></p> Since there can be multiple ways to correct a sentence, this assesses whether the correction is unnecessarily verbose or makes unnecessary syntax changes. The best correction should be done with the minimum number of edits. For example, if the sentence <em>"I wentt at Rome for my birthday"</em> is corrected to <em>"I decided to go to Rome for my birthday"</em> this should be penalized under this category because it contains unnecessary syntax changes, even though the final sentence is grammatically correct. This metric answers the question: Is the system over-correcting or making unnecessary syntax changes? The answers should be "No", "Minor over-correction", "Moderate over-correction" or "Substantial over-correction".</li> <div style="height: 15px;" /> <div style="height: 10px;" />  </ul></ul> <strong>NOTE: please take a screeshot of this page for your future reference as you may not be able to access it again once you start the survey.</strong>
> ```
>
> - GPT4 PROMPT FOR SIMPLIFICATION
> ```
>  You're GPT4 and are about to start a task where you will be shown some pieces of text taken mostly from older news articles, alongside 4 different possible simplified versions of each text, and you will be asked to evaluate the quality of the simplifications based on some metrics defined below. The purpose of text simplification can be varied, but include things such as making the text easier to read, more accessible to language learners, or explaining a medical or legal text in plain speak. We ask that you carefully read the original text and rank each of the 4 simplifications according to the following metrics, which are defined below.
>
>
> Semantics, or adequacy: This assesses whether the meaning of the text is preserved further to the simplification. Semantic preservation is assessed on a 5-point Likert scale from 1 (Meaning Not Preserved) to 5 (Meaning fully preserved). NOTE: You should penalize simplified text which is not self contained. For example, if the sentence "Scientists have no idea how eels protect their own brain and muscles from being damaged by electricity" is simplified as "Scientists have no idea how they stay safe", this should be rated around 2 out of 5 as it is not possible by just reading the simplified version to understand what "they" is referring to. Like wise if the simplified text contains hallucinations (facts not contained in the source) it should be penalised. For example if "He yelled" is simplified as "His father Russell shouted" this should be rated around 2 under this category. The omission of some unimportant details from the source into the simplified version is acceptable at your discretion, if the text meaning is preserved.
>
>
> Grammaticality (or fluency). This assesses whether the simplified text remains grammatical and understandable. Sentences should have no formatting problems, capitalization errors or obviously ungrammatical sentences (e.g., fragments, missing components) that make the text difficult to read. Grammaticality / Fluency is assessed on a 5-point Likert scale from 1 (Not Fluent) to 5 (Super Fluent)
>
>
> Simplicity. This assesses whether the simplified text, considered in its entirety, is simpler than the source text. Simplicity is assessed on a 5-point Likert scale from 1 (text is significantly more complex than the source) to 5 (text is significantly simpler than the source, or no further simplification was possible). Note that sometimes you may encounter very short sentences which cannot be simplified further and are repeated identically, for example "How are you?". In such cases, even if the text is not simpler than the source, it may be appropriate to assign a score of 5. A score of 3 is appropriate for cases where the text was not simplified nor made more complex, but potential for simplifcation exists.
>
>
> We will pass you the input you need to rank in json format
> Please reply with the scores in json format.
> This is an example json query where "original_input" is the source text, 'id' is the unique identifier, and all other keys represent output texts which you need to evaluate.
> {"original_input": "He is the first to market a simplified version of the tool to the public , a project that , for now , is not dangerous .", "gold_reference": "Zayner is the first to sell a gene-changing tool to regular people . His set is not dangerous .", "google/flan-t5-xxl": "He is the first to market a simplified version of the tool to the public, a project that, for now, is not dangerous.", "davinci-instruct-beta": "He is the first to market a simplified version of the tool to the public.\n\nHe is the first to market a simplified version of the tool to the public.", "gpt-3.5-turbo": "He created a safer version of the tool and introduced it to the public before anyone else.", "id": "9ec83328fbea94d7cdc4b92aba92f64abaa8727f6c0c99cc3b588b58bb8bb3f5"}
>
>
> Your answer should contain the id and the scores, for example, if you wish to give gold_reference a semantics score of 2, a grammaticality score of 5 and a simplicity score of 4, and you wish to give google/flan-t5-xxl a semantics score of 5, a grammaticality score of 5, a simplicity score of 4, and you wish to give davinci-instruct-beta a semantics score of 5, a grammaticality score of 5, a simplicity score of 4, and you wish to give gpt-3.5-turbo a semantics score of 5, a grammaticality score of 5 and a simplicity score of 5, then you should return the following output (note how the id item needs to be preserved to allowed for identification):
> {"gold_reference": {"semantics": 2, "grammaticality": 5, "simplicity": 4}, "google/flan-t5-xxl": {"semantics": 5, "grammaticality": 5, "simplicity": 4}, "davinci-instruct-beta": {"semantics": 5, "grammaticality": 5, "simplicity": 4}, "gpt-3.5-turbo": {"semantics": 5, "grammaticality": 5, "simplicity": 5}, "id": "9ec83328fbea94d7cdc4b92aba92f64abaa8727f6c0c99cc3b588b58bb8bb3f5"}
>
>
> Is this clear? Do you have any questions or are you ready to start?
> ```
>
> - GPT4 PROMPT FOR SUMMARIZATION
> ```html
> You're GPT4 and are about to start a task where you will be shown some pieces of text taken mostly from older news articles, alongside 4 different possible summaries of each text, and you will be asked to evaluate the quality of the summaries based on some metrics defined below. The purpose of text summarization is to make the text shorter and more concise without changing its overall meaning, omitting unimportant details while retaining the key content of the story. We ask that you carefully read the original text and rank each of the 4 summaries according to the following metrics, which are defined below.
>
>
> Relevance. This assesses the quality of selection of important content from the source. The summary should include only important information from the source document. Please penalize summaries which contain redundancies and excess information. Relevance is assessed on a 5-point Likert scale from 1 (Not Relevant) to 5 (Super Relevant)
>
>
> Fluency. This assesses the quality of individual sentences. Sentences in the summary should have no formatting problems, capitalization errors or obviously ungrammatical sentences (e.g., fragments, missing components) that make the text difficult to read. Fluency is assessed on a 5-point Likert scale from 1 (Not Fluent) to 5 (Super Fluent)
>
>
> Coherence. This assesses the collective quality of all sentences. The summary should be well-structured and well-organized. The summary should not just be a heap of related information, but should build from sentence to sentence to a coherent body of information about a topic. Coherence is assessed on a 5-point Likert scale from 1 (Not Coherent) to 5 (Super Coherent)
>
>
> Consistency. This assesses the factual alignment between the summary and the summarized source. A factually consistent summary contains only statements that are entailed by the source document. Summaries that contain hallucinated facts (information which is not present in the source document) should be penalized. Consistency is assessed on a 5-point Likert scale from 1 (Not Consistent) to 5 (Super Consistent)
>
>
> We will pass you the input you need to rank in json format.
> Please reply with the scores in json format.
> This is an example json query where "original_input" is the source text, "id" is the unique identifier, and all other keys represent output texts which you need to evaluate.
> {"original_input": "(CNN)Love it or hate it, Jared Leto's interpretation of the Joker is an internet sensation. The Oscar winner put on white makeup (and a lot of tattoos this time) to portray the Clown Prince of Crime in the upcoming movie \"Suicide Squad.\" Set for release August 5, 2016, \"Suicide Squad\" is based on the DC Comics series and also stars Will Smith, Margot Robbie and Viola Davis. Twitter users got their first look at Leto in character Friday night, and the memes started almost immediately. From comparisons to \"Home Alone\" to an imagining of Ben Affleck tatted up, people on social media put their Photoshopping skills to work all weekend. Which is your favorite?", "gold_reference": "Leto will play the Clown Prince of Crime in 2016's \"Suicide Squad\"\nThe first picture of Leto in character led to a series of spoof photos .", "bigscience/T0pp": "The internet is buzzing about Jared Leto's new look as the Joker. The Oscar winner is playing the Clown Prince of Crime in \"Suicide Squad\"", "text-davinci-003": " Jared Leto's interpretation of the Joker in the upcoming movie \"Suicide Squad\" has become an internet sensation. The movie, set for release August 5, 2016, also stars Will Smith, Margot Robbie and Viola Davis. Twitter users got their first look at Leto in character Friday night, and the memes started almost immediately, with people Photoshopping him in various ways.", "gpt-3.5-turbo": "Jared Leto's portrayal of the Joker in the upcoming movie \"Suicide Squad\" has become an internet sensation. Twitter users have been sharing memes and Photoshopped images of Leto's character, who is heavily tattooed, since his first appearance in character on Friday. The movie is set for release on August 5, 2016 and also stars Will Smith, Margot Robbie, and Viola Davis.", "id": "49f45c17ee34e2aa420ef557d944a6f2df96f5226c2793e4a9d379c517fde9a5"}
>
>
> Your answer should contain the id and the scores, for example, using the example given above, if you wish to give gold_reference a relevance score of 2, a fluency score of 5, a coherence score of 4 and a consistency score of 4, and you wish to give bigscience/T0pp a relevance score of 3, a fluency score of 5, a coherence score of 1 and a consistency score of 3,  and you wish to give text-davinci-003 a relevance score of 5, a fluency score of 4, a coherence score of 3 and a consistency score of 4, and you wish to give gpt-3.5-turbo a relevance score of 4, a fluency score of 3, a coherence score of 5 and a consistency score of 3, then you should return the following output (note how the id item needs to be preserved to allow for identification):
> {"gold_reference": {"relevance": 2, "fluency": 5, "coherence": 4, "consistency": 4}, "bigscience/T0pp": {"relevance": 3, "fluency": 5, "coherence": 1, "consistency": 3}, "text-davinci-003": {"relevance": 5, "fluency": 4, "coherence": 3, "consistency": 4}, "gpt-3.5-turbo": {"relevance": 4, "fluency": 3, "coherence": 5, "consistency": 3}, "id": "49f45c17ee34e2aa420ef557d944a6f2df96f5226c2793e4a9d379c517fde9a5"}
>
>
> Is this clear? Do you have any questions or are you ready to start?
> ```
>
> - GPT4 PROMPT FOR GRAMMATICAL ERROR CORRECTION
> ```html
> You're GPT4 and are about to start a task where you will be shown some sentences written by learners of English. Some of these sentences will contain errors, and alongside each sentence you will be shown 4 different possible corrections, and you will be asked to evaluate the quality of the correction based on some metrics defined below. This task is called Grammatical Error Correction (GEC), and is the task of automatically detecting and correcting errors in text. The task not only includes the correction of grammatical errors, such as missing prepositions and mismatched subject-verb agreement, but also orthographic and semantic errors, such as misspellings and word choice errors respectively. Note that not all sentences you will see include grammatical errors; if they do not, we would expect the corrected version to be identical to the source. We ask that you carefully read the original sentence and rank each of the 4 corrections according to the following metrics, which are defined below.
>
>
> Semantics. This assesses whether the meaning of the text is preserved following the GEC. Semantic preservation is assessed on a 5-point Likert scale from 1 (Meaning Not Preserved) to 5 (Meaning fully preserved). NOTE: You should penalize corrections which change the meaning unnecessarily. For example, the sentence "I wentt at Rome for my birthday" should be corrected to "I went to Rome for my birthday". A correction such as "I went to Rome for my anniversary" should be penalised in this category as it introduces unnecessary changes to the meaning.
>
>
> Grammaticality. This assesses the quality of the correction and answers the question "How many errors are left in the corrected sentence?". Please provide a count of the remaining errors, regardless of whether they were present in the source or they were newly introduced errors in the supposed corrected version. The options are "0", "1", "2 or more". Note that, unlike for semantics where a score of 5 is better than a score of 1, here a score of "0" is better than a score of "1" which is better than a score of "2 or more" (this is because if there are 0 errors remaining, the GEC task has been fulfilled).
>
>
> Over-correction. Since there can be multiple ways to correct a sentence, this assesses whether the correction is unnecessarily verbose or makes unnecessary syntax changes. The best correction should be done with the minimum number of edits. For example, if the sentence "I wentt at Rome for my birthday" is corrected to "I decided to go to Rome for my birthday" this should be penalized under this category because it contains unnecessary syntax changes, even though the final sentence is grammatically correct. This metric answers the question: Is the system over-correcting or making unnecessary syntax changes? The answers should be "No", "Minor over-correction", "Moderate over-correction" or "Substantial over-correction".
>
>
> We will pass you the input you need to rank in json format.
> Please reply with the scores in json format.
> This is an example json query where "original_input" is the source sentence, "id" is the unique identifier, and all other keys represent the output corrected sentences which you need to evaluate.
> {"original_input": "Travalling by car is also much more convenient.", "gold_reference": "Travelling by car is also much more convenient.", "opt-iml-max-30b": "Travalling by car is also much more convenient.", "text-davinci-003": "Traveling by car is also much more convenient.", "gpt-3.5-turbo": "Traveling by car is also much more convenient.", "id": "765d0adf8a8c58e2617a435be0d2451edcfcb782492ac662432da53843607d9a"}
>
>
> Your answer should contain the id and the scores, for example, using the example given above, if you wish to give gold_reference a semantics score of 5, a grammaticality score of "0", an overcorrection score of "No", and you wish to give opt-iml-max-30b a semantics score of 4, a grammaticality score of "1", an overcorrection score of "Minor over-correction",  and you wish to give text-davinci-003 a semantics score of 3, a grammaticality score of "2 or more", an overcorrection score of "Substantial over-correction", and you wish to give gpt-3.5-turbo a semantics score of 5, a grammaticality score of "0", an overcorrection score of "Moderate over-correction", then you should return the following output (note how the id item needs to be preserved to allow for identification):
> {"gold_reference": {"semantics": 5, "grammaticality": "0", "overcorrection": "No"}, "opt-iml-max-30b": {"semantics": 4, "grammaticality": "1", "overcorrection": "Minor over-correction"}, "text-davinci-003": {"semantics": 3, "grammaticality": "2 or more", "overcorrection": "Substantial over-correction"}, "gpt-3.5-turbo": {"semantics": 5, "grammaticality": "0", "overcorrection":  "Moderate over-correction"}, "id": "49f45c17ee34e2aa420ef557d944a6f2df96f5226c2793e4a9d379c517fde9a5"}
>
>
> Is this clear? Do you have any questions or are you ready to start?
> ```

---

### Official Review · Reviewer_EMpQ · 2023-08-04

**Soundness:** 3

**Excitement:**

3: Ambivalent: It has merits (e.g., it reports state-of-the-art results, the idea is nice), but there are key weaknesses (e.g., it describes incremental work), and it can significantly benefit from another round of revision. However, I won't object to accepting it if my co-reviewers champion it.

**Paper Topic And Main Contributions:**

In this paper the authors perform a thorough evaluation of LLMs for three NLP tasks: text summarization, text simplification, and grammatical error correction. They use both automatic metrics and human evaluations based on scores.
They also experiment with using GPT-4 as if it were a "human evaluator". The findings are that ChatGPT outperforms other models according to human reviewers while scoring poorly when using automatic metrics; that the gold reference of popular benchmarks are scored worse than the output models; and that GPT-4 shows a reasonable performance as an evaluator (based on the Krippendorff alpha coefficient), although its agreement with the human raters is always lower. The code and datasets used are public and available.

**Questions For The Authors:**

- One thing that was not clear from my reading of the paper is if all the annotators (including GPT) were given the same set of results to evaluate, or if it was different subsets of the test data. I understand the outputs were normalized to fit a ranking from 1 to 4, but did they all annotate the same results? In that case, why did you use Krippendorff alpha instead of Fleiss kappa, that would be a good fit for many raters labeling the same data?

- It is not clear why you did not include in Table 1 the automatic scores for every evaluated open-source model. I think it is related to the outputs taken into consideration for human evaluation (Table 2). Can you clarify that please?

- You mention that GPT-4 might be evaluating input-output pairs that were used for training it. Did you consider the possibility of designing a few new test examples for each dataset, in order to tackle this problem and obtain a stronger conclusion about this?

- In the ethics statement you consider that this research does not increase the risk of harm caused by these models. However, some people might find the conclusion of using GPT-4 as an evaluator a bit risky. What are your thoughts about systematic LLM-based evaluations are performed for new models in the future?

- As far as I understand, the "Semantics" category used in GEC human evaluation is introduced by you. However it seems to overlap with the "Over-correction" criterion, as described in appendix C using "I wentt at Rome for my Birthday" as an example. This can be a deeper problem when asking annotators to distinguish between the two criteria, making the annotation even harder. Can you clarify the differences between the two?

EDIT: The authors clarified these questions, some of these points should be included in the paper.

**Reasons To Accept:**

The paper presents an interesting approach to reflect about LLM evaluation. NLP practitioners might find aspects of this work useful, since the evaluation dilemma is becoming stronger on a daily basis.

This work can also trigger interesting discussions, for example about the usage of LLMs to evaluate other NLP models or systems or how they perform in relation to automatic metrics or human evaluators.

**Reasons To Reject:**

Although this work is really interesting for the community, it could be much better with a deeper analysis of the presented experiments. For example, one of the objectives of this work is to evaluate how GPT-4 correlates with human evaluators. However, the agreement is not reported using pairs (annotator1 and annotator 2, annotator 2 and GPT-4, etc.) but aggregating all the annotators hence not allowing further discussion. Since they note that GPT-4 does not correlate perfectly with human evaluators, the agreement can also be analyzed depending on the range of human scores (e.g. what is the agreement between annotator 1 an GPT 4 when annotator 1 gives a 2 score vs when gives a 5 score).

Furthermore, as the authors indicate, there could be an important bias regarding model selection based on automatic metrics, which means that human evaluators were not able to evaluate outputs from models that were discarded in the first evaluation round.

**Reproducibility:**

4: Could mostly reproduce the results, but there may be some variation because of sample variance or minor variations in their interpretation of the protocol or method.

**Reviewer Confidence:**

4: Quite sure. I tried to check the important points carefully. It's unlikely, though conceivable, that I missed something that should affect my ratings.

---

> ### Author Rebuttal · Authors · 2023-08-29
>
> We thank Reviewer 1 for the useful comments.  Please see below our responses to their main concerns and questions.
>
> - **Reason to Reject 1**: Regarding the lack of reported agreement using annotator pairs (annotator1 and annotator 2, annotator 2 and GPT-4, etc.), we note that we added very detailed annotator-by-annotator results in Table 6 within Appendix E. We weren’t able to include these in the main corpus due to page limitations. While the Krippendorff scores are not reported in Table 6, they can easily be calculated using the results from the Table. Moreover, the Krippendorff scores in Table 2 are split into `alpha_1` and `alpha_2`, where `alpha_1` includes the three annotators and `alpha_2` includes the three annotators and GPT4, so we can already clearly see that the three annotators have a higher agreement with each other than with GPT4 overall; however we are very happy to include annotator pairs Krippendorff coefficients within Appendix E for the camera-ready version. Regarding us not reporting the agreements based on the range of human scores (e.g. what is the agreement between annotator 1 an GPT 4 when annotator 1 gives a 2 score vs when gives a 5 score), we felt that this was outside the main scope of the paper. Moreover, since we only analysed 100 samples using human reviewers and GPT-4, we felt it was not a large enough sample size to provide meaningful insights when looked at such a granular level of detail; for example, a reviewer might consistently give a score of 4 for a standard that a different reviewer considers worth of a score of 5, and as such we felt that looking at the correlation between annotators’ average rankings for each model was more relevant than focusing on individual score-by-score agreement.
> - **Reason to Reject 2**: The reviewer raised another valid point regarding human evaluators not being able to evaluate outputs from models that were discarded in the first evaluation round, potentially introducing bias. We do understand this concern, and as human evaluation is very time intensive and costly, we had to narrow down the scope of this experiment; however, we would like to reassure the reviewer that we (the authors) manually looked at these models’ outputs ourselves before making a decision on whether it was sensible to use automatic metrics for selection of models to be evaluated during human evaluation, and came to the conclusion this was the best course of action.
> - **Question 1**: We can confirm that all three human annotators and GPT-4 annotated the exact same set of 100 samples for each of the three tasks. We apologise for the confusion and will make that clearer in the camera-ready version. While both Fleiss kappa and Krippendorff alpha are suitable for handling many (>2) raters, we chose to use Krippendorff alpha for a number of reasons: 1) Krippendorff seems to be reported more often in similar literature, and in particular we decided to follow (Fabbri et al., TACL 2021)[1] in reporting Krippendorff alpha, 2) while Fleiss kappa can be applied to ordinal (ranked) data, we found it was more commonly used when assessing binary or categorical ratings, 3) Krippendorff alpha is more general and also suited to handle missing data, unlike Fleiss kappa (we didn’t have missing data in our case, but this may explain why we saw it reported more frequently across the literature we reviewed). In the end, both Fleiss kappa and Krippendorff alpha would be suitable and sensible coefficients to use in this case, and we just made a decision to use the latter.
>
> [1] Alexander R. Fabbri, Wojciech Kryściński, Bryan McCann, Caiming Xiong, Richard Socher, and Dragomir Radev. 2021. SummEval: Re-evaluating Summarization Evaluation. Transactions of the Association for Computational Linguistics, 9:391–409.
> - **Question 2**: This was mostly due to space constraints, as we experimented with multiple prompts and temperature values for each model and only reported the best combination for each in Table 1, reporting all combinations would have taken up the whole page, and, as the reviewer correctly identified, consistency with Table 2 was another reason for this choice, as we only asked human reviewers to review a subset of the models based on automatic metric results, due to time and budgetary constraints. However, we do report the full set of automatic scores for every evaluated open-source and closed-source model in Tables 3, 4 and 5 within Appendix D.
> - **Question 3**: This is a good and relevant question. We didn’t create new data for evaluation as we made the decision to stick to open-source widely used datasets. We also felt that only adding a few samples created by us might not lead to strong confidence in our conclusions, and might also affect reproducibility. Moreover, generating a substantial amount of high-quality new samples for evaluation (at least 100 input-output pairs per task) would be very time and cost intensive and therefore outside the scope of this work. However, it is certainly an interesting proposition which we could leave for future research work.
> - **Question 4**: LLM-based evaluations certainly have the potential of reducing the need for human evaluators, so this is a very valid observation. However, at the moment traditional automatic evaluation metrics are still widely used (and they also don’t require human evaluators), and as we’ve shown, they’re clearly inadequate at evaluating the latest generation of LLMs. Therefore there is a desperate need for better evaluation strategies. Considering the costs of employing human evaluators, it’s very unlikely that they will ever entirely replace automatic evaluation, and as such there is a strong case to be made in exploring LLMs to replace automatic rule-based evaluation metrics. We also believe the market will ultimately and inevitably adjust to any new technology as it’s been the case so far, and if LLMs can reach human-level judgement in the future (which is still not yet the case), then evaluators’ time could possibly be freed for more complex and engaging tasks.
> - **Question 5**: We agree that an over-correction could lead to the meaning (semantics) being changed. As such, there could be some overlap between these two metrics, and you could reasonably argue that most corrections leading to a change of meaning, are also over-corrections. However, the reverse is not necessarily the case, for example, changing a sentence from “I wentt at Rome for my birthday” to “I decided to go to Rome for my birthday” doesn’t significantly affect the meaning of the sentence, but it nonetheless represents a clear case of over-correction. As such we felt there would be value in keeping them separate to get a better insight into whether and how frequently over-corrections change the meaning of the sentence or not. We are more than happy to clarify this difference in the camera-ready version.

---

### Meta-Review · Area_Chair_ztwg · 2023-09-18

**Recommendation:** 4

**Metareview:**

The authors of this paper evaluate generative LLMs on three NLP tasks, namely text summarization, text simplification, and grammatical error correction, using both automatic metrics and human evaluations. They also experiment with using GPT-4 as if it were a "human evaluator".

The reviewers highlighted the following reasons to accept:
- a systematic evaluation of multiple LLMs on three tasks, leading to clear and important takeaways: automatic evaluation metrics are insufficient for evaluating LLM outputs; GPT-4 is a good evaluator; ground truth references are bad
- interesting negative result that GPT4 is not very good as an evaluator for grammar error correction

The following reasons to reject were mentioned:
- A more detailed analysis of inter-annotator agreement would have been desirable. The authors noted that some of the requested information is available in an appendix, and that other parts of the requested discussion are either not possible or not meaningful with the available data.
- Lacking details of the experimental settings leading to poor reproducibility. The authors gave complementary information in the rebuttal that addressed most issues. One reviewer decided to keep a low reproducibility score due to the confusing and changing OpenAI model naming conventions, which are however totally beyond the control of this paper's authors.
- Limited novelty, in the sense that similar conclusions about human vs. automatic evaluations have been drawn before. The authors responded that there is no prior work on text simplification or grammatical error correction evaluation in connection with LLMs.
- Some limitations are not presented sufficiently prominently. The authors promised to make the requested changes in the camera-ready version.

---

### Decision · Program_Chairs · 2023-10-07

**Decision:**

Accept-Main

**Comment:**

The authors of this paper evaluate generative LLMs on three NLP tasks, namely text summarization, text simplification, and grammatical error correction, using both automatic metrics and human evaluations. They also experiment with using GPT-4 as if it were a "human evaluator".

The reviewers highlighted the following reasons to accept:
- a systematic evaluation of multiple LLMs on three tasks, leading to clear and important takeaways: automatic evaluation metrics are insufficient for evaluating LLM outputs; GPT-4 is a good evaluator; ground truth references are bad
- interesting negative result that GPT4 is not very good as an evaluator for grammar error correction

The following reasons to reject were mentioned:
- A more detailed analysis of inter-annotator agreement would have been desirable. The authors noted that some of the requested information is available in an appendix, and that other parts of the requested discussion are either not possible or not meaningful with the available data.
- Lacking details of the experimental settings leading to poor reproducibility. The authors gave complementary information in the rebuttal that addressed most issues. One reviewer decided to keep a low reproducibility score due to the confusing and changing OpenAI model naming conventions, which are however totally beyond the control of this paper's authors.
- Limited novelty, in the sense that similar conclusions about human vs. automatic evaluations have been drawn before. The authors responded that there is no prior work on text simplification or grammatical error correction evaluation in connection with LLMs.
- Some limitations are not presented sufficiently prominently. The authors promised to make the requested changes in the camera-ready version.